# Going beyond gadgets: the importance of scalability for analogue quantum simulators

Dylan Harley [1] ✉, Ishaun Datta [2], Frederik Ravn Klausen[1], Andreas Bluhm [3], Daniel Stilck França [4], Albert H. Werner [1] & Matthias Christandl[1]

Quantum hardware has the potential to efficiently solve computationally difficult problems in physics and chemistry to reap enormous practical rewards. Analogue quantum simulation accomplishes this by using the dynamics of a controlled many-body system to mimic those of another system; such a method is feasible on near-term devices. We show that previous theoretical approaches to analogue quantum simulation suffer from fundamental barriers which prohibit scalable experimental implementation. By introducing a new mathematical framework and going beyond the usual toolbox of Hamiltonian complexity theory with an additional resource of engineered dissipation, we show that these barriers can be overcome. This provides a powerful new perspective for the rigorous study of analogue quantum simulators.

The simulation of quantum systems has long been identified as a potential application for quantum technologies[1], for which long-term benefits may range from condensed matter physics to quantum chemistry and the life sciences[2,3]. This problem is classically intractable, owing to exponential growth in the number of parameters required to describe the state of a many-body system, whereas the advantage of quantum hardware for this purpose is obvious: one merely has to prepare the required many-body state. On a universal quantum computer, time evolution can then be discretised and approximated by a quantum circuit, through a series of quantum gates[4]. This approach, known as digital quantum simulation[5], has seen extensive theoretical development[6,7] and remains a promising route towards attaining quantum advantage[8]. However, useful and scalable simulations remain out of reach for near-term technology[9] due to the requirement of a large universal fault-tolerant quantum computer. In this work, we focus on an alternative approach: analogue quantum simulation.

Broadly speaking, an analogue quantum simulator consists of an engineered and well-controlled many-body system with adjustable interactions, with the capability to prepare initial states and perform measurements[10]. By tuning such a system, one aims to mimic a different target system; in this way, computing the dynamics of the target system can be accomplished through the native time evolution of the simulator, without requiring the application of a universal set of gates.

These more modest physical requirements promise near-term potential for analogue quantum simulation, despite the inherent limitations fixed by a given experimental apparatus.

Characterisation of analogue quantum simulators is, unlike the digital case, relatively under-explored from a theoretical perspective. Existing work in this direction includes that of Cubitt et al.[11], in which the authors define a notion of Hamiltonian simulation in terms of low-energy encodings: a low-energy subspace of the simulator Hamiltonian is required to approximate the spectrum of the target Hamiltonian. This notion has been extraordinarily successful in making complexity-theoretic reductions between various Hamiltonians, leading to the classification of many so-called universal families[12–16] which have the power to simulate all of many-body physics. Such reductions do not necessarily aim to capture experimental possibilities, however: as we prove, the relatively simple task of encoding a system of $n$ non-interacting qutrits into a linear number of qubits in this regime ends up requiring a simulator system whose individual interactions scale as $\Omega(n)$. This scaling arises due to the dimension mismatch when one encodes a qutrit into a set of qubits, resulting in unwanted local configurations that must be prohibited in the low-energy subspace by a large energy penalty (see Fig. 1). Similar scalings are observed to arise through the use of Hamiltonian gadgets[17,18], a tool used for many Hamiltonian reductions. Although the qutrit-to-qubit result does not extend to the case where the $n$ qutrits are simulated by $\Omega(n^2)$ qubits, we

[1]Department of Mathematical Sciences, University of Copenhagen, Universitetsparken 5, 2100 Copenhagen, Denmark. [2]Stanford University, 450 Serra Mall, Stanford, CA 94305, USA. [3]Univ. Grenoble Alpes, CNRS, Grenoble INP, LIG, 38000 Grenoble, France. [4]Univ. Lyon, ENS Lyon, UCBL, CNRS, Inria, LIP Lyon Cedex 07, France. ✉e-mail: dh@math.ku.dk

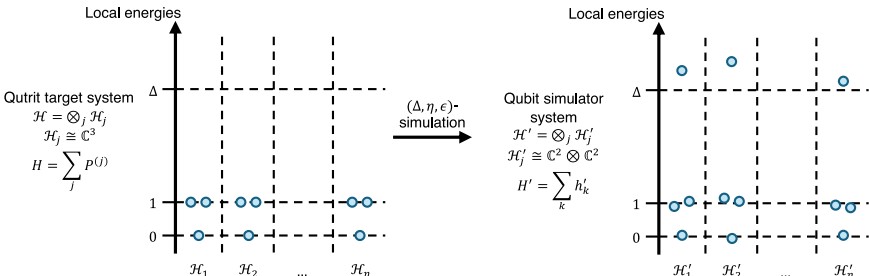

**Fig. 1 | Qutrit-to-qubit encoding energies.** A sketch of the on-site energies for a system of non-interacting qutrits under a low-energy encoding, before and after simulation in the sense introduced by Cubitt et al.[11]. In this example, each qutrit is mapped to a system of two qubits. The original Hamiltonian consists of a sum $\sum_j P^{(j)}$ of rank two projectors $P^{(j)}$ applied to each qutrit, resulting in energy levels 0,1,1. In order to simulate the qutrit Hamiltonian, an energy penalty of at least $\Delta$ must be given to one of the four local states at each site. This is a simplification: in general the simulator sites $\mathcal{H}'_j$ may interact and hence local energies are not well-defined.

also note that blowing up the system size may, in some cases, necessitate strong interactions in order for correlations to spread fast enough through the enlarged system.

We argue that for practical applications to large-scale many-body simulations, such as for quantum chemistry, the simulation of an $n$-site many-body system should not require the implementation of individual interactions whose magnitude scales with $n$. The necessity of this requirement is clear from a logistical perspective since an experimental device will only be able to implement a bounded range of energy scales for a single interaction. However, there is also philosophical motivation to be suspicious of such scalings: a many-body Hamiltonian is inherently a modular object, and an analogue quantum simulation should reflect this. The addition of a few qubits and local interactions to one end of the physical system should require an analogous action on the simulator—it should not require the adjustment of every other interaction in the system.

Despite this additional requirement of scalability, there is also a sense in which the framework of[11] can be relaxed: the requirement to simulate the full physics of a target Hamiltonian in the low-energy subspace of a simulator is unnecessary in many cases. For example, one may only wish to simulate a specific set of local observables, or exploit symmetries to restrict to an invariant subspace under the Hamiltonian (a regime explored, in the case of a low-energy subspace, by Aharonov et al.[19]). Furthermore, as the experimental distinction between analogue and digital devices becomes increasingly blurred[9,20], it is important to consider a range of experimental possibilities beyond pure Hamiltonian evolution, such as intermediate unitary pulses and open-system dynamics.

In this work, we propose a mathematical framework for analogue quantum simulators to address the above points and capture the full scope of experimentally realisable systems. We additionally develop a general characterisation for Hamiltonian gadgets, and find rigorous no-go results for their scalable use for locality reduction. Finally, we construct a new dissipative gadget that circumvents the restrictions we find in the pure Hamiltonian case. In the regime of scalable quantum simulators, we do not expect to talk about a simple class of universal Hamiltonians that can simulate all others in any sense resembling previous results. On the other hand, the more general notion of simulation, which we outline in this work, gives rise to a new notion of universality, not phrased in terms of Hamiltonian classification but rather the dynamics of observables. We expect that here a resource theory of simulation should arise, with the power of simulators related by a partial order in analogy to the theory of multipartite states and tensor networks[21–23].

## Results

### The analogue quantum simulator
In this section, we describe our mathematical framework for analogue quantum simulation, for which further details can be found in the

Methods section. The capabilities of a simulator are characterised by a target Hamiltonian $H$, a set of states $\Omega_{state}$, and a set of observables $\Omega_{obs}$; the goal of the simulator is then to approximate the evolution of the observables in $\Omega_{obs}$ under $H$, starting from initial states in $\Omega_{state}$. Restricting the set of states $\Omega_{state}$ may offer practical and theoretical benefits: for example, one could reduce to those that can be reliably prepared on an experimental device, or take advantage of the symmetries of $H$ to restrict $\Omega_{state}$ to a specific invariant subspace and simulate only the reduced Hamiltonian. Likewise, $\Omega_{obs}$ may reflect the capabilities of the measurement apparatus, or for a many-body system one might take advantage of a highly localised set of observables to only simulate their Lieb-Robinson light cone[24], significantly reducing the hardware overhead necessary to simulate for small times. Such techniques have been studied in the context of many-body state exploration[25,26], and more recently in the realm of analogue simulation[27].

An analogue quantum simulator can be mathematically described in terms of three components, which we illustrate in Fig. 2. First, a state encoding $\mathcal{E}_{state}$, which maps initial states from the target set $\rho \in \Omega_{state}$ into the simulator system. This is defined in terms of a quantum channel, allowing one to interpret the target state as a quantum input to the simulator, in contrast to the regime of fault-tolerant digital quantum computers whose input is ultimately classical. Next, the simulator's time evolution is specified by a family of quantum channels $\{T_t\}_{t\in[0,t_{max}]}$. These describe the dynamics of the simulator, for example the evolution under a simulator Hamiltonian $H'$. However, one could also consider $T_t$ accounting for interactions with a bath (such capability is required by the criteria for analogue simulators given by Cirac et al.[10]), modelling errors, or capturing other engineered controls reflecting the possibilities of the experimental apparatus. The final component of the simulator is an encoding for observables $\mathcal{E}_{obs}$, a unital and completely positive map which sends an observable of interest $O \in \Omega_{obs}$ to the relevant observable to be measured on the simulator system.

After the encoding and time evolution steps for a given initial state $\rho \in \Omega_{state}$ and time $t \in [0, t_{max}]$, the simulator system lies in the state $(T_t \circ \mathcal{E}_{state})(\rho)$, upon which one measures the encoded observable $\mathcal{E}_{obs}(O)$ for a chosen $O \in \Omega_{obs}$. The simulation has accuracy $\epsilon$ if the expectation value of this measurement is within $\epsilon$ of its target value— that is, the expected value of measuring $O$ on $e^{-itH}\rho e^{itH}$. Note that rather than a Hamiltonian $H$, one could just as easily consider the simulation of an open target system, for instance described by a quantum dynamical semigroup[28,29].

For practical simulators, some constraints must be placed on the maps used in this definition. We generally assume that both $\mathcal{E}_{state}$ and $\mathcal{E}_{obs}$ are local in a sense we define, to ensure that local errors correspond to local noise on the target system, and that local observables can be measured locally on the simulator system. Moreover, the time evolution channel $T_t$ should be implementable without the need for

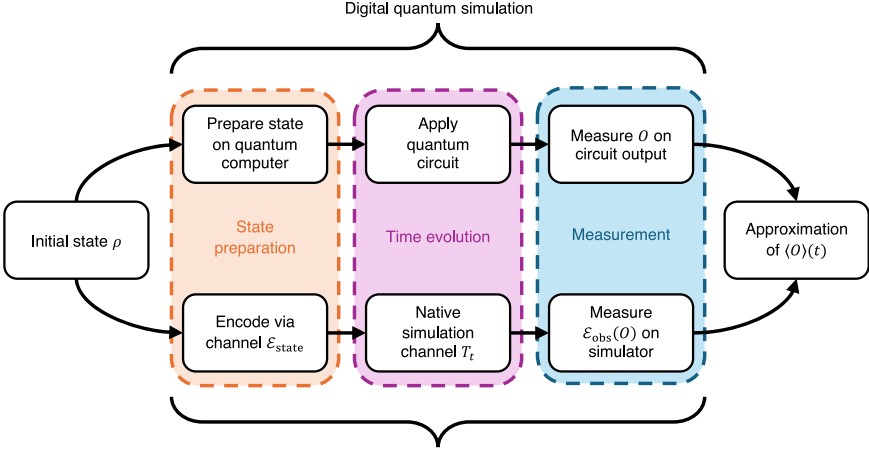

**Fig. 2 | Analogue and digital simulation.** A schematic description of our framework for analogue quantum simulation, in contrast with the digital approach. Both approaches aim to compute the observable expectation value $\langle O \rangle(t) = \text{tr}[Oe^{-itH}\rho e^{itH}]$, given an initial state $\rho$. The analogue simulator, which uses state encodings $\mathcal{E}_{\text{state}}$ and $\mathcal{E}_{\text{obs}}$ respectively, has accuracy $\epsilon$ if $|\text{tr}[Oe^{-itH}\rho e^{itH}] - \text{tr}[\mathcal{E}_{\text{obs}}(O)(T_t \circ \mathcal{E}_{\text{state}})(\rho)]| \leq \epsilon$.

feed-forward measurement results for adaptive control: the lack of error correction is an important and characteristic feature of analogue simulators.

## Generalising Hamiltonian gadgets

A ubiquitous technique for Hamiltonian reductions in complexity theory is the use of so-called Hamiltonian gadgets[17,18,30,31]. These provide a recipe to simulate complicated many-body interactions from a more restrictive family, for example, to simulate a 3-body interaction using 2-body interactions. In this section, we arrive at a general formalism for such constructions, in order to prove that they are associated with unavoidable energy scalings which pose a significant challenge for experimental realisations. The usual procedure, as formulated by Bravyi et al.[18] for example, is as follows: starting from a target Hamiltonian $H$ (which might be a single interaction in a far larger system), one first adjoins an ancillary qubit $m$. On this enlarged system one defines the gadget Hamiltonian by $H' = \Delta |1\rangle\langle 1|_m + V$, where $\Delta \gg 1$ is a large parameter used to define a low-energy subspace approximately in terms of the $|0\rangle$ state of the mediator qubit, and $V$ is a relatively small term which, via a perturbative approximation, effectively simulates the target $H$ in this low-energy subspace.

It is not surprising that this method of construction generically requires strong interactions corresponding to the large value of $\Delta$ needed to provide a sufficiently high energy penalty, but it is not immediately clear that there is no way around this cost (possibly outside of the perturbative regime). Indeed, several works[32–34] have explored the optimisation of Hamiltonian gadgets for practical implementation, though generally the problem scaling interactions is not eliminated entirely. In this work, we produce a generalized framework for gadgets in order to prove a lower bound for such scalings, suggesting that such techniques may be unsuitable for experiments on large systems. Our results are summarised in Fig. 3b, and full mathematical details can be found in the Methods section.

Let $H$ be the target Hamiltonian on a Hilbert space $\mathcal{H}$, and let the gadget Hamiltonian $H'$ act on the space $\mathcal{H} \otimes \mathcal{A}$, for $\mathcal{A}$ some ancillary system. We require the following two properties of $H'$, illustrated in Fig. 3a:

- Accuracy: The spectrum of $H$ should be approximated by that of $H'$, in some subspace defined by a projector $P' \in \text{Proj}(\mathcal{H} \otimes \mathcal{A})$, up to error $\epsilon \geq 0$.
- Combination: The above property should hold even when any additional Hamiltonian $H_{\text{else}}$ is added to both $H$ and $H'$, at the

expense of an additional spectral error $\zeta \|H_{\text{else}}\|$, where $\| \cdot \|$ denotes the operator norm.

For small error parameters $\epsilon$ and $\zeta$, these properties are—nontrivially—sufficient to force $H'$ to satisfy the following definition, which resembles previously studied notions of simulation[11,18]. We say that $H'$ is an $(\eta, \epsilon)$-gadget (where $\eta$ is a new parameter related to $\zeta$, also measuring the ability of the gadget to combine with other terms) for $H$ if there exists a projector $P \in \text{Proj}(\mathcal{A}) \setminus \{0\}$ and a unitary operator $U \in U(\mathcal{H} \otimes \mathcal{A})$ satisfying two conditions. Firstly, $U$ must be $\eta$-close to the identity: $\|U - \mathbb{I}\| \leq \eta$. Then, defining the projector $P'$ by $P' = U(\mathbb{I} \otimes P)U^\dagger$ (so that $\eta$ in some sense quantifies how close $P'$ is to a pure projection on the ancillary system), the second condition ensures that the spectrum $P'H'P'$ should approximate that of $H$, up to some multiplicity: $\| P'H'P' - U(H \otimes P)U^\dagger \| \leq \epsilon$.

This gadget definition expresses the quality of a gadget through two parameters: $\epsilon$ can be thought of as the absolute error of the gadget, whilst $\eta$ bounds the error incurred when the gadget is combined with other interactions in a Hamiltonian. In particular, when $\|H_{\text{else}}\| \sim n$ grows with the size of the system, $\eta$ must correspondingly shrink to hold the error constant.

Despite the generality of this definition, it is sufficient to guarantee that such gadgets can be combined in parallel. That is, given a many-body Hamiltonian $H = \sum_i H_i$ and sufficiently good gadgets $H_i'$ for each of the individual terms $H_i$, the Hamiltonian $H' = \sum_i H_i'$ constitutes a good gadget for $H$. A similar result holds for low-energy gadgets (for which the projector $P'$ is replaced by a projector onto the low-energy subspace of $H'$), and also leads to a generalisation of the ground state energy estimation result of Bravyi et al.[31].

On the other hand, we show that, when used for certain types of reduction, gadgets come at an unavoidable energy cost. In particular, any attempt to simulate a $k$-body interaction $H$ via a gadget $H'$ consisting of $k'$-body interactions for $k' < k$ necessarily requires interaction strengths scaling as $\Omega(\eta^{-1})$. In order to control the absolute error of a many-body system, $\eta^{-1}$ must grow with the size of the system, leading to unfeasible energy scalings and constituting a significant barrier for Hamiltonian reductions in the regime of experimentally realisable analogue quantum simulators.

## Gadgets from the quantum Zeno effect

To circumvent our no-go result for scalable Hamiltonian locality reduction, in this section, we exhibit a new kind of gadget, taking advantage of the non-unitary possibilities afforded by a general

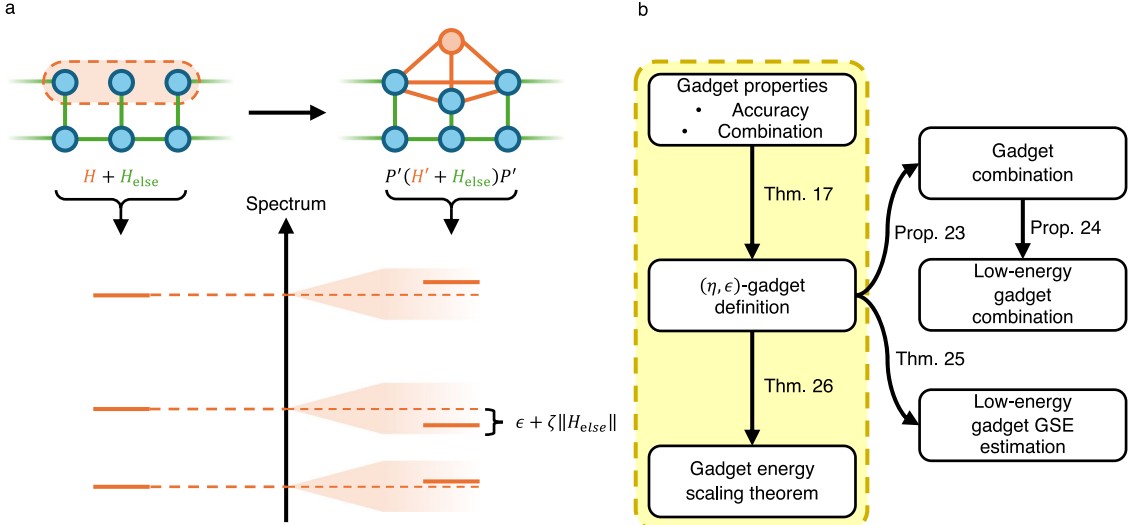

**Fig. 3 | Hamiltonian gadget characterisation. a** The interaction hypergraph of a Hamiltonian containing a 3-local interaction which is replaced by a 2-local gadget. The gadget property requires that the spectrum is unchanged up to $\epsilon + \zeta \|H_{\text{else}}\|$, for $\epsilon, \zeta > 0$, when restricted by a projector $P'$. **b** Structure of gadget results; boxes highlighted in yellow indicate the central argument for the energy scaling no-go.

We first formalise the desirable properties of gadgets and show that they imply a general definition, from which we can prove the energy scaling theorem along with various combination properties, including a generalisation of a result of Bravyi et al.[31] for ground state energy (GSE) estimation.

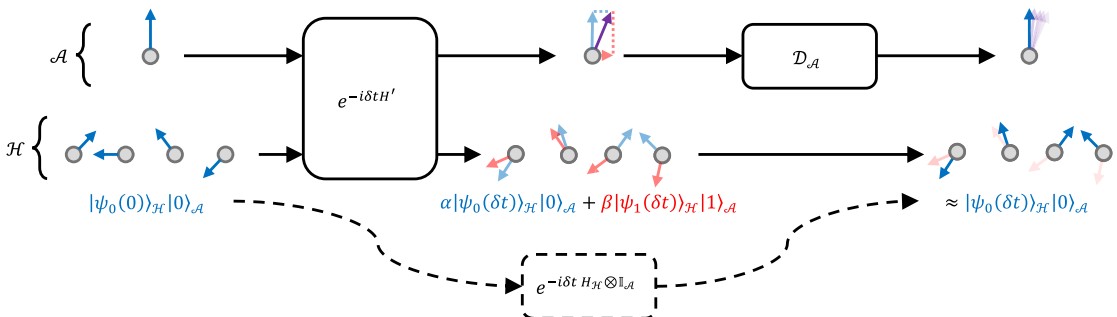

**Fig. 4 | Dissipative gadget evolution.** Circuit representation of the non-unitary gadget procedure for a single timestep. The initial state $|\psi_0(0)\rangle_{\mathcal{H}} |0\rangle_{\mathcal{A}}$ is evolved under a Hamiltonian $H'$ for time $\delta t$, resulting in a superposition of states with the ancillary qubit in the $|0\rangle_{\mathcal{A}}$ and $|1\rangle_{\mathcal{A}}$ positions. After applying the dissipative channel to the $\mathcal{A}$ system, the system collapses to its $|0\rangle_{\mathcal{A}}$ state with high probability due to the quantum Zeno effect. Meanwhile, the resulting state on the $\mathcal{H}$ system approximately corresponds to evolution under a different Hamiltonian, $H_{\mathcal{H}}$.

simulation channel $T_t$. This works by restricting the mediator qubit to its $|0\rangle$ state not with a strong interaction but through inertia induced by the quantum Zeno effect. This powerful resource can be used to build a direct $k$-to-$(\lceil k/3 \rceil + 1)$-local gadget, without interactions scaling with the size of the system.

The recipe for this construction is qualitatively similar to that for usual Hamiltonian gadgets: starting from a target interaction $H$, a mediator qubit $m$ is adjoined to the system, and evolves under a simulator Hamiltonian $H'$. In this case, however, $H'$ need not contain an interaction $\Delta|1\rangle\langle 1|_m$, with $\Delta$ scaling with the size of the system. Instead, a dissipative channel is applied to the qubit $m$ at regular intervals separated by time $\delta t$. Provided that $\delta t$ is small enough, the quantum Zeno effect keeps $m$ effectively fixed in its $|0\rangle$ state with high probability, whilst the remainder of the system evolves as though under the target Hamiltonian $H$. This is illustrated by Fig. 4, and a rigorous description can be found in the Methods.

These non-unitary gadgets may be combined with other terms in a Hamiltonian at no extra cost (effectively corresponding to a gadget with combination error parameter $\eta = 0$), yielding an improvement on any possible pure Hamiltonian gadget. On the other hand, the construction has various caveats: strong interactions (though not scaling with system size) are still necessary for high accuracy of a single

gadget, and we expect that combining multiple such gadgets will require strong interactions, to suppress the probability of any ancillary qubit transitioning into the $|1\rangle$ state. Moreover, the precisely engineered stroboscopic dissipation channel constitutes a new experimental challenge.

Nevertheless, this construction provides insight into the ways in which non-unitary dynamics might be exploited for practical analogue quantum simulation problems—indeed, similar tools have already found applications in theory[35,36] and in practice[37] for digital quantum computing. In light of our theorems implying extensive interaction scaling for qutrit-to-qubit mappings and gadget locality reduction, which effectively serve as no-go theorems for practical universal simulators built from pure Hamiltonian dynamics, we anticipate that similar hybrid techniques will constitute a powerful tool for attaining useful quantum advantage with quantum simulators.

## Methods
### Criteria for quantum computation and simulation
In a review of the prospective possibilities of quantum computing[38] the author provided a set of requirements, now known as the DiVincenzo criteria, designed to serve as a full specification for implementations of universal quantum computers. These are summarised in Fig. 5.

| The DiVincenzo criteria for quantum computation | | The Cirac-Zoller criteria for quantum simulation | |
|---|---|---|---|
| A. | A scalable physical system with well-characterised qubits. | I. | A system of bosons/fermions confined in a region in space, containing a large number of degrees of freedom. |
| B. | The ability to initialise the state of the qubits to a simple fiducial state, such as \|000 ... ⟩. | II. | The ability to approximately prepare a known quantum state – ideally pure. |
| C. | Long relevant decoherence times, much longer than the gate operation time. | III. | The ability to engineer and adjust the values of a set of interactions between the particles, and possibly external fields or a reservoir. |
| D. | A universal set of quantum gates. | IV. | The capability to measure the system, either on specific sites or collectively. |
| E. | A qubit-specific measurement capability. | V. | A procedure for increasing confidence in the results of the simulation. |

**Fig. 5 | A summary of the DiVincenzo[38] and Cirac-Zoller[10] criteria.** The DiVincenzo criteria provide necessary and sufficient requirements for universal digital quantum computers. Similarly, the Cirac-Zoller criteria offer a set of requirements for analogue quantum simulation, for which universality may not be available.

As well as concretely providing the experimentalist with a necessary set of criteria to aim towards, the sufficiency of the DiVincenzo criteria provides the theorist with a canonical yardstick to judge the applicability of their protocol to idealised quantum hardware. It is, therefore, important that such requirements reflect exactly what can be expected from quantum technology in the long term, neither excluding feasible technologies nor including unfeasible procedures.

A similar set of criteria for analogue quantum simulators is discussed by Cirac et al.[10], also summarised in Fig. 5. These are all natural requirements to ask of a quantum simulator, but it is noteworthy that criterion III does not provide any restriction on the interactions that one should expect the simulator to include. This leads to a problem which does not arise for the DiVincenzo criteria: whereas a quantum computer can approximate arbitrary $k$-qubit gates from the compact set $U((\mathbb{C}^2)^{\otimes k})$ of unitary transformations relatively cheaply due to the Solovay-Kitaev theorem[39], the task of an analogue quantum simulator is to implement $k$-qudit interactions from the unbounded set of possible Hamiltonians $\text{Herm}((\mathbb{C}^d)^{\otimes k})$. The ability to realise arbitrarily strong interactions on a physical device is clearly an impossibility.

Thus, the key extra criterion which we demand of an analogue quantum simulator is that the encoding of the target Hamiltonian should be size-independent. Concretely, if the Hamiltonian $H$ to be encoded consists of local interactions $(h_i)_{i=1}^m$ on $n$ sites then the encoding of individual terms should not depend, for instance by polynomial scaling of interaction strengths, on the size of the physical system $n$. In particular, we argue that methods for practical analogue quantum simulation must respect a limit on the interaction strengths of the simulator Hamiltonian. The strongest interactions should be bounded by some constant fixed by physical limitations, and the weakest interactions should be similarly bounded from below (since sufficiently weak interactions will be overwhelmed by noise in the simulator). In addition, in order to ensure the local and size-independent encoding of each site into the simulator, we argue that the simulator should grow no faster than linearly with the size of the target system. If each site is encoded into more than $O(1)$ simulator sites, it will be impossible to encode the full system into a simulator of the same dimension while preserving geometric locality (without introducing scaling interactions). We summarise these requirements with the following qualitative definition:

**Definition 1.** (Size-independent simulation) We say that an analogue quantum simulation is size-independent if the simulation of a $n$-site Hamiltonian can be implemented scalably with $n$. By this, we mean that the number of qubits used in the simulation should grow no faster than

linearly in $n$, and the interaction strengths necessary should remain $\Theta(1)$.

It is worth noting that further formalisation is required to make this definition robust. For example, suppose we are given a Hamiltonian $H = h_1 + h_2$ where $\|h_1\|, \|h_2\| = O(n^{-1})$, which violates the size-independence requirement. One could simply define $h_1' = h_1 + K$, $h_2' = h_2 - K$, for some $K = \Theta(1)$, and then $H = h_1' + h_2'$ can be written in a form which does not obviously violate Definition 1. To exclude such possibilities, we could impose an additional requirement that $H$ is given in a canonical form, such as that described by Wilming et al.[40].

As well as being experimentally and qualitatively desirable, encoding interactions independently has quantitative benefits; as noted by Cubitt et al.[11], for a suitably local Hamiltonian encoding, local errors on the simulator system will correspond to local errors on the target system. For NISQ hardware, this represents an extremely useful way to mitigate the negative effects of a noisy simulation: rather than random scrambling, noise can be viewed as the manifestation of physically reasonable noisy effects on the target system.

Finally, studying the power of Hamiltonians subject to interaction energies that are constant in system size is well-motivated in its own right, from the perspective of Hamiltonian complexity. For example, Aharonov et al.[19] show that restriction to such Hamiltonians will necessarily sacrifice some sense of the universality of the simulator. Earlier results in Hamiltonian complexity theory[31], however, show that in many cases, it is still possible to simulate ground state energies up to an extensive error.

## Hamiltonian complexity theory

We say that a Hamiltonian $H$ on the space of $n$ qubits $\mathcal{H} = (\mathbb{C}^2)^{\otimes n}$ is $k$-local if it can be written as $H = \sum_{j=1}^N h_j$, where each of the terms $h_j$ acts on at most $k$ of the qubit sites. We consider the $h_j$ individual interactions in the Hamiltonian and make reference to the interaction hypergraph, whose vertices are qubits and whose (hyper)edges are interactions (joining the qubits on which they act), illustrated in Fig. 6.

Informally, the $k$-local Hamiltonian problem asks whether the ground state energy of a $k$-local Hamiltonian is less than $a$, or greater than $b$, for some real numbers $a < b$ separated by a suitably large gap. This problem lies in the QMA complexity class: the natural quantum analogue to the classical NP, containing problems whose solutions can be efficiently verified (but not necessarily found) on a quantum computer.

**Definition 2.** ($k$-local Hamiltonian problem) The $k$-local Hamiltonian problem is the promise problem which takes as its input a $k$-local Hamiltonian $H = \sum_{j=1}^N h_j$ on the space of $n$ qubits $\mathcal{H} = (\mathbb{C}^2)^{\otimes n}$, where

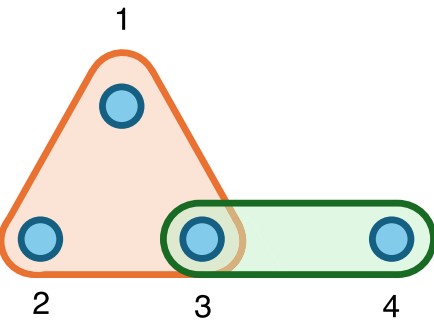

$$H = A_{123} \otimes \mathbb{I}_4 + \mathbb{I}_{12} \otimes B_{34}$$

**Fig. 6 | Example Hamiltonian interaction hypergraph.** A Hamiltonian $H$ on 4 qubits, and its associated interaction hypergraph. The Hamiltonian consists of a 3-local (orange) and a 2-local (red) term, so we say that $H$ is 3-local.

$N = \text{poly}(n)$, and for each $j$ we have $\|h_j\| \le \text{poly}(n)$ and $h_j$ is specified by $O(\text{poly}(n))$ bits.

Given $a < b$ with $b - a > 1/\text{poly}(n)$, let $\lambda_0(H)$ denote the lowest eigenvalue of $H$. Then the output should distinguish between the cases

- Output 0: The ground state energy of $H$ has $\lambda_0(H) \le a$.
- Output 1: The ground state energy of $H$ has $\lambda_0(H) \ge b$.

Through the Feynman-Kitaev circuit-to-Hamiltonian construction[41], it was established that the 5-local Hamiltonian problem is QMA-complete, and subsequent works optimising the construction[30] and using gadget techniques[42] reduced this further to show the QMA-completeness of the 2-local Hamiltonian problem. Various further optimisations have been found to refine the problem and further restrict the family of allowed Hamiltonians (see for example, refs. 43,44); indeed hardness results have been shown to hold even under the significant restriction to 1-dimensional translationally invariant systems[45]. QMA-completeness is closely related to a notion of universality for simulators; an equivalence was proved by Kohler et al.[15].

The constructions involved in the aforementioned results contain Hamiltonian interaction strengths which scale polynomially, or exponentially, with system size—such Hamiltonians are infeasible for an analogue simulator. A notable exception to this is the work of Bravyi et al.[31], in which the authors use the Schrieffer-Wolff transformation to show that bounded-strength interactions are sufficient for one to reproduce the ground state energy of the original Hamiltonian up to an extensive error.

As much of this Hamiltonian simulation literature focuses on specific complexity-theoretic problems, comparatively little work has been done to actually define a mathematical framework for analogue quantum simulation to be used in experiment. Notable recent work in this direction includes that of Cubitt et al.[11], in which the authors study methods of encoding Hamiltonians via a map $\mathcal{E}_{\text{obs}} : \text{Herm}(\mathcal{H}) \to \text{Herm}(\mathcal{H}')$, which satisfy the natural requirement of preserving the spectrum of observables. Additionally, in the case that $\mathcal{H} = \otimes_{i=1}^{n} \mathcal{H}_i$ is a space of many sites, they introduce the further notion of local encodings, which map local observables in $\mathcal{H}$ to local observables in $\mathcal{H}' = \otimes_{i=1}^{n} \mathcal{H}_i'$. By deriving the most general possible form of a spectrum-preserving Hamiltonian encoding, and then imposing natural locality conditions, the authors arrive at the following definition.

**Definition 3.** (Local Hamiltonian encoding[11]) A local Hamiltonian encoding is a map $\mathcal{E}_{\text{obs}} : \text{Lin}(\otimes_{i=1}^{n} \mathcal{H}_i) \to \text{Lin}(\otimes_{i=1}^{n} \mathcal{H}_i')$ of the form

$$\mathcal{E}_{\text{obs}}(M) = V(M \otimes P + \bar{M} \otimes Q)V^{\dagger}, \tag{1}$$

where $P$ and $Q$ are locally distinguishable orthogonal projectors on an ancillary space $\mathcal{A} = \otimes_{i=1}^{n} \mathcal{A}_i$, and $V = \otimes_{i=1}^{n} V_i$ where $V_i \in \text{Isom}(\mathcal{H}_i \otimes \mathcal{A}_i, \mathcal{H}_i')$ for all $i$. Here $\bar{M}$ denotes the complex conjugate of the matrix $M$.

Projectors $P, Q \in \text{Proj}(\otimes_i \mathcal{A}_i)$ are locally distinguishable if, for all $i$, there exist orthogonal projectors $P_i, Q_i \in \text{Proj}(\mathcal{A}_i)$ such that $(P_i \otimes \mathbb{I})P = P$ and $(Q_i \otimes \mathbb{I})Q = Q$. Generally, we consider the case of $\text{rank}(P) > 0$ (referred to as standard[11]), for which one can define a corresponding state encoding

$$\mathcal{E}_{\text{state}}(\rho) = V(\rho \otimes \tau)V^{\dagger}, \tag{2}$$

where $\tau$ is a state on $\mathcal{A}$ satisfying $P\tau = \tau$.

Moreover, the authors define the following notion of simulation, which relaxes the requirements of locality and allows for some error in the simulated eigenvalues.

**Definition 4.** ($(\Delta, \eta, \epsilon)$-simulation[11]) A Hamiltonian $H' \in \text{Herm}(\mathcal{H}') = \text{Herm}(\otimes_{i=1}^{n} \mathcal{H}_i')$ is said to $(\Delta, \eta, \epsilon)$-simulate a Hamiltonian $H \in \text{Herm}(\mathcal{H}) = \text{Herm}(\otimes_{i=1}^{n} \mathcal{H}_i)$ if there exists a local encoding (Definition 3) $\mathcal{E}_{\text{obs}}(M) = V(M \otimes P + \bar{M} \otimes Q)V^{\dagger}$ such that

(i) There exists an encoding $\tilde{\mathcal{E}}_{\text{obs}}(M) = \tilde{V}(M \otimes P + \bar{M} \otimes Q)\tilde{V}^{\dagger}$ (where $\tilde{V} \in \text{Isom}(\mathcal{H} \otimes \mathcal{A}, \mathcal{H}')$ need not have a tensor product structure as in Definition 3) such that $\| V - \tilde{V} \| \le \eta$ and $\tilde{\mathcal{E}}_{\text{obs}}(\mathbb{I}) = P_{\le \Delta(H')}$ is the projection onto the low-energy ($\le \Delta$) subspace of $H'$, and

(ii) $\| P_{\le \Delta(H')} H' P_{\le \Delta(H')} - \tilde{\mathcal{E}}_{\text{obs}}(H) \| \le \epsilon$.

This approach (later generalised by Apel et al.[46] and refined with resource constraints by Zhou et al.[16]) provides an elegant framework to capture a notion of one Hamiltonian fully simulating another. However, we believe that this regime does not capture the scope of possibilities for analogue quantum simulation experiments. On one hand, the formalism requires the entire physics of the target system to be encoded into the low-energy subspace of a simulator—this rules out simulators which only simulate part of the target system, or in a different subspace. On the other hand, the formalism is too broad in the sense that it does not prohibit unrealistically scaling interaction strengths in violation of Definition 1.

**Framework**

The generic task of an analogue quantum simulator is to estimate the dynamics of observables in a system $\mathcal{H}$ under the evolution of a target Hamiltonian $H$, up to some maximum time $t_{\text{max}}$. In particular, it is not always necessary to simulate the entire target system in arbitrary configurations: it may be convenient to restrict to a particular subset of initial states $\Omega_{\text{state}}$, for example lying in a subspace invariant under the Hamiltonian or corresponding to the states which can be reliably prepared by the simulator, and similarly to a particular subset of observables of interest $\Omega_{\text{obs}}$. We denote by $\mathcal{H}'$ the Hilbert space corresponding to the simulator system, and for $t \in [0, t_{\text{max}}]$ we write $T_t : D(\mathcal{H}') \to D(\mathcal{H}')$ for the family of time evolution quantum channels implemented by the simulator, where $D(\mathcal{H}')$ is the set of density matrices on $\mathcal{H}'$. This approach, in which we view simulations in terms of individual observables rather than the entire Hamiltonian, has been considered in earlier work[25-27].

The minimal requirement for a simulator is that it should approximate the expectation values of the elements of $\Omega_{\text{obs}}$. That is, $\text{tr}[Oe^{-iHt}\rho e^{iHt}]$ should be close to $\text{tr}[O'T_t(\rho')]$ for all $\rho \in \Omega_{\text{state}}$ and $O \in \Omega_{\text{obs}}$, where $\rho'$ and $O'$ are some encoded versions of the states and operators respectively. Notice that, in principle, the experimentalist could be using a completely different simulator for each choice of $\rho$ and $O$, with $\mathcal{H}'$ a space large enough to contain all of them and by encoding $\rho$ into several copies. However, this would violate the size-independence requirement of Definition 1. if $\Omega_{\text{obs}}$ and $\Omega_{\text{state}}$ both do not only contain $O(1)$ elements. Furthermore, it is natural to consider analogue quantum simulators as machines taking quantum, rather than classical, input—possibly prepared by another experiment—which

cannot be cloned. For this reason, we assume that the state encoding takes the form of a quantum channel $\mathcal{E}_{\text{state}} : D(\mathcal{H}) \to D(\mathcal{H}')$. Correspondingly, to accommodate for quantum outputs, we require the observable encoding $O \mapsto O'$ to be a unital and completely positive map $\mathcal{E}_{\text{obs}} : \text{Herm}(\mathcal{H}) \to \text{Herm}(\mathcal{H}')$. This ensures that the Hilbert-Schmidt dual operator $\mathcal{E}^*_{\text{obs}}$ is a quantum channel, so measurement of $\mathcal{E}_{\text{obs}}(O)$ on $\rho'$ can equivalently be thought of as a measurement of $O$ on a decoded state $\mathcal{E}^*_{\text{obs}}(\rho)$. This perspective sets analogue quantum simulators apart from the framework of digital quantum computation, for which fault-tolerant architectures require both inputs and outputs to be classical.

This definition is still sufficiently versatile to capture the simulation of global observables that are a sum of local parts $O = \sum_k O_k$ (a task, for example, useful for variational quantum algorithms[47]), in the following way. Often the $O_k$ cannot be simultaneously measured due to non-commutativity relations or experimental limitations. The simplest approach to estimating $O$ is to run many simulations, measuring one of the $O_k$ each time (this process can be sped up by combining simultaneously measurable terms[48]), and summing the average results.

The above discussion leads us to the following definition, which is illustrated in Fig. 2.

**Definition 5.** (Analogue quantum simulation) Given a set of states $\Omega_{\text{state}}$ on a Hilbert space $\mathcal{H}$, a normalised set of observables $\Omega_{\text{obs}}$ (i.e. $\|O\| = 1$ for all $O \in \Omega_{\text{obs}}$, where $\|\cdot\|$ denotes the operator norm), a time $t_{\text{max}} > 0$, a Hamiltonian $H \in \text{Herm}(\mathcal{H})$, and $\epsilon > 0$, we say that a family of quantum channels $T_t : D(\mathcal{H}') \to D(\mathcal{H}')$, for $t \in [0, t_{\text{max}}]$ simulates $H$ with respect to $\Omega_{\text{state}}$ and $\Omega_{\text{obs}}$ with accuracy $\epsilon$ if there exists

1. A state encoding quantum channel $\mathcal{E}_{\text{state}} : D(\mathcal{H}) \to D(\mathcal{H}')$ which maps states to the simulator Hilbert space $\mathcal{H}'$,
2. An observable encoding, given by a unital and completely positive map $\mathcal{E}_{\text{obs}} : \text{Herm}(\mathcal{H}) \to \text{Herm}(\mathcal{H}')$,

such that

$$|\text{tr}[\mathcal{E}_{\text{obs}}(O)(T_t \circ \mathcal{E}_{\text{state}})(\rho)] - \text{tr}[O(e^{-itH} \rho e^{itH})]| \leq \epsilon, \qquad (3)$$

for all $\rho \in \Omega_{\text{state}}$, $O \in \Omega_{\text{obs}}$, and $t \in [0, t_{\text{max}}]$.

Our use of a Hamiltonian $H$ for the target system is mostly for simplicity; the simulation of more general dynamics, of open quantum systems, for example, can be defined analogously, with the target Hamiltonian $H$ replaced by any generator of a quantum dynamical semigroup[28,29]. It should be noted also that Definition 5 could equivalently have been phrased in terms of a set of POVMs rather than observables $\Omega_{\text{obs}}$. We use the latter for convenience in relating our work to other results. It is plausible that one could engineer a time-dependent observable observable encoding $\mathcal{E}_{\text{obs}}$, but here we restrict our focus to the time-independent case to avoid the complexity of the simulation task being hidden in this step.

By the triangle inequality, (3) holds for any convex combination of the states and observables in $\Omega_{\text{state}}$ and $\Omega_{\text{obs}}$ respectively, so we could without loss of generality assume that the two sets are convex to begin with.

Often the simulation channels $T_t$ in Definition 5 are taken simply as time evolution under some simulator Hamiltonian $H' \in \text{Herm}(\mathcal{H}')$, but it is useful to consider a more general case. Firstly, this allows one to directly account for, and possibly exploit, dissipative errors in the experimental setup[49]. Secondly, it enables the possibility of a more complicated simulation experiment, for example involving intermediate measurements. Moreover, it is important to allow the simulation of open quantum systems for our definition to be consistent with criterion III of Fig. 5. Despite the generality afforded by Definition 5, we emphasise that experimentally practical simulations should be size-independent as in Definition 1. That is, the implementation of $T_t$ should not require engineering a system of size, which grows more than linearly in $n$, or boundlessly scaling interaction energies. Another

important constraint is that $T_t$ should not include the use of adaptive channels based on feed-forward measurements—hence distinguishing the process from digital quantum computation.

We note that Hamiltonian models of quantum computation such as quantum walks[50] and previous notions of dynamical Hamiltonian simulation[51] are not consistent with our definition of analogue simulation: such constructions also incur scalings in both the system size and in necessary evolution time (corresponding to scalings in interaction strength, if time is normalised) which violate the size-independence conditions of Definition 1.

### Local encodings

Although Definition 5 is phrased in terms of general encoding maps, it is practically useful to ensure that states and observables are encoded in a way that is both practical to implement and behaves favourably with respect to noise. In this section, we present such a notion of local encodings and state some basic properties; proofs are contained in Supplementary Note 1. A similar discussion is presented for the stronger case of local Hamiltonian encodings by Cubitt et al.[11], and a discussion of the stability of local observable measurements to local noise is given by Trivedi et al.[27].

**Definition 6.** (Local state encoding) Let $\mathcal{H} = \otimes_{i=1}^{n} \mathcal{H}_i$ and $\mathcal{H}' = \otimes_{j=1}^{n'} \mathcal{H}'_j$. We say that a state encoding $\mathcal{E}_{\text{state}} : D(\mathcal{H}) \to D(\mathcal{H}')$ is local if it has a Stinespring representation of the form

$$\mathcal{E}_{\text{state}}(\rho) = \text{tr}_E[U(\rho \otimes |0\rangle\langle0|_F)U^\dagger], \qquad (4)$$

where $F = \otimes_k F_k$ and $E = \otimes_l E_l$ are ancillary systems and $U \in \text{U}(\mathcal{H} \otimes F, \mathcal{H}' \otimes E)$ is a constant-depth quantum circuit.

It is immediate that constant-depth quantum circuits (built from one-qubit and two-qubit gates) preserve locality. That is, given a local operator $A$ on $\mathcal{H} \otimes F$, the operator $UAU^\dagger$ is local (acting on the forward light cone of the support of $A$) on $\mathcal{H}' \otimes E$, and similarly for the inverse $U^\dagger$. In fact, it is known in the theory of quantum cellular automata that this constraint is equivalent to representability as a constant-depth quantum circuit[52].

For simulating physical systems, one particularly desirable feature of a simulator is local error back-propagation. That is, local noise on the simulator system should correspond in some way to local (perhaps realistic) noise on the target system. Ideally, we would like to prove that for any state $\rho \in D(\mathcal{H})$ and local error channel $\mathcal{N}' : D(\mathcal{H}') \to D(\mathcal{H}')$ on the simulator, there exists a corresponding local error channel $\mathcal{N} : D(\mathcal{H}) \to D(\mathcal{H})$ on the target system satisfying

$$\mathcal{N}' \circ \mathcal{E}_{\text{state}}(\rho) \stackrel{?}{=} \mathcal{E}_{\text{state}} \circ \mathcal{N}(\rho). \qquad (5)$$

However, we cannot hope to prove this in general, since the noise operator $\mathcal{N}'$ may take the simulator system outside the image of $\mathcal{E}_{\text{state}}$. Instead we have a slightly weaker version of this statement, which is a direct consequence of the causal structure of local state encodings.

**Proposition 7.** (Local error back-propagation) Let $\mathcal{E}_{\text{state}} : D(\mathcal{H}) \to D(\mathcal{H}')$ be a local state encoding as in Definition 6, and let $\mathcal{N}' : D(\mathcal{H}') \to D(\mathcal{H}')$ be a channel whose Kraus operators $\{X'_k\}$ each act on $O(1)$ sites in $\mathcal{H}'$. Then there exists a channel $\mathcal{N} : D(\mathcal{H} \otimes F) \to D(\mathcal{H} \otimes F)$ whose Kraus operators $\{X_k\}$ each act on $O(1)$ sites in $\mathcal{H} \otimes F$, and such that for all $\rho \in \mathcal{H}$,

$$\mathcal{N}' \circ \mathcal{E}_{\text{state}}(\rho) = \text{tr}_E[U \mathcal{N}(\rho \otimes |0\rangle\langle0|_F)U^\dagger]. \qquad (6)$$

In other words, local noise on the simulator corresponds to local noise on the target system and ancillary encoding system. The corresponding result with locality replaced by geometric locality holds in

the case when the light cones of $U$ are local with respect to the underlying geometry of the simulator and target systems.

Similarly, we have local forward-propogation under such an encoding, in the sense that local operations on a site $\mathcal{H}_i$ to $\rho \in D(\mathcal{H})$ will not affect the reduced density matrix $\mathrm{tr}_A[\mathcal{E}_{\mathrm{state}}(\rho)]$, where $A$ is the forward light cone of $\mathcal{H}_i$ under $U$ in $\mathcal{H}'$.

We define local observable encodings analogously to the state encoding case.

**Definition 8.** (Local observable encoding) Let $\mathcal{H} = \otimes_{i=1}^n \mathcal{H}_i$ and $\mathcal{H}' = \otimes_{j=1}^{n'} \mathcal{H}'_j$. We say that an observable encoding $\mathcal{E}_{\mathrm{obs}} : \mathrm{Herm}(\mathcal{H}) \to \mathrm{Herm}(\mathcal{H}')$ is local if it is the adjoint (with respect to the Hilbert-Schmidt inner product) of a local state encoding.

It is immediate from this definition that one can measure the encoded observable $\mathcal{E}_{\mathrm{obs}}(O)$ by first applying the constant-depth quantum circuit $\mathcal{E}^*_{\mathrm{obs}} : D(\mathcal{H}') \to D(\mathcal{H})$ to $\rho' \in D(\mathcal{H}')$, and then measuring $O$. When $O$ is local, we can alternatively implement the measurement via a local POVM directly on the simulator system.

**Proposition 9.** (Encoded measurements) Let $\mathcal{E}_{\mathrm{obs}}$ be a local observable encoding as in Definition 8, and let $O$ be a local operator on $\mathcal{H}$. Then $\mathcal{E}_{\mathrm{obs}}(O)$ can be measured using a local POVM on $\mathcal{H}'$.

### Applications of the framework

In this section, we discuss some basic applications of our notion of analogue quantum simulation in the sense we have introduced in Definition 5. Firstly, we give an example of a trivial but illustrative situation in which encoding qudits into qubits incurs an unavoidable cost for low-energy encodings, but which is not an issue in our framework. We then demonstrate the robustness of the definition under noise, and show that it is consistent with the existing notion of simulation given in Definition 4. Finally, we note how Lieb-Robinson bounds can be used to reduce the overhead of simulating local observables.

**Qudits to qubits**. To motivate this example, we first notice that the requirement of Cubitt et al.[11] (Definition 4) that the simulator Hamiltonian should reproduce the target dynamics in its low-energy subspace is too strong for some practical situations. As observed by the authors, this can require the simulator to use strong interactions to push unwanted states out of the low-energy subspace. Proposition 10 provides a formal statement of this fact (proved in Supplementary Note 2) in the context of encoding a simple qutrit Hamiltonian into qubits.

Here we consider qutrits with individual state spaces $\mathbb{C}^3$ spanned by a basis $\{|\downarrow\rangle, |0\rangle, |\uparrow\rangle\}$. We write $P_0^{(j)} = |0\rangle\langle 0|$ and $P_\uparrow^{(j)} = |\uparrow\rangle\langle\uparrow|$, where the superscript indicates that the projectors act on the $j$th qutrit.

**Proposition 10.** Let $\mathcal{H} = (\mathbb{C}^3)^{\otimes n}$ be the space of $n$ qutrits acted on by the Hamiltonian

$$H_n = \sum_{j=1}^n (P_0^{(j)} + P_\uparrow^{(j)}). \tag{7}$$

Suppose $H'_n = \sum_{j=1}^K h'_j$ is a $k$-local Hamiltonian on $\mathcal{H}' = (\mathbb{C}^2)^{\otimes m}$, where $m = O(n^{1+\alpha})$, for $\alpha \geq 0$ and $k = O(1)$. Assume the interaction hypergraph of $H'_n$ has degree bounded by $d = O(1)$.

If $H'_n$ is a $(\Delta, \eta, \epsilon)$-simulation for $H_n$ in the sense of Definition 4, for $\eta \in [0, 1)$ and $\epsilon \geq 0$, then

$$\max_j \| h'_j \| = \Omega(n^{1-\alpha}(1 - \eta^2)). \tag{8}$$

From (8) we see that simulating this simple system with a low-energy encoding, an interaction hypergraph of bounded degree, and bounded locality, requires either the qubit count or interaction energy

(or a mixture) to scale unfeasibly with $n$. This constitutes a violation of the requirements of Definition 1. and imposes an unnecessary experimental requirement for the task of simulating non-interacting qutrits. The proof of this fact follows from a dimension-counting argument, since the state space of the qutrits cannot be surjectively encoded into the qubit simulator, see Fig. 1. In contrast, the simulation task is trivial in our framework given in Definition 5 because the low-energy encoding requirement is relaxed.

Letting $H_n = \sum_{j=1}^n (P_0^{(j)} + P_\uparrow^{(j)})$ as in Proposition 10, we can simulate all observables under $H_n$ on $\mathcal{H}' = \otimes_{j=1}^n (\mathbb{C}^2 \otimes \mathbb{C}^2)$ via any isometry

$$V : \mathbb{C}^3 \to \mathbb{C}^2 \otimes \mathbb{C}^2, \tag{9}$$

encoding each qutrit into two qubits. To realise a simulator in the sense of Definition 5, we let

$$\mathcal{E}_{\mathrm{state}} : \rho \mapsto V^{\otimes n} \rho (V^{\otimes n})^\dagger, \quad \mathcal{E}_{\mathrm{obs}} : O \mapsto V^{\otimes n} O (V^{\otimes n})^\dagger, \tag{10}$$

and

$$T_t = e^{-it\mathcal{E}_{\mathrm{obs}}(H_n)}(\cdot) e^{it\mathcal{E}_{\mathrm{obs}}(H_n)}, \tag{11}$$

which is just time evolution under a 2-local Hamiltonian with bounded-strength interactions.

Although Proposition 10 does not necessarily rule out simulations in which the $n$ qutrits are encoded into $\Omega(n^2)$ qubits, such approaches suffer from a different problem. Generally, if each qudit in a $D$-dimensional system is encoded into $\Omega(n^\alpha)$ qudits for $\alpha > 0$, whilst keeping the dimension fixed, then the inflated system size will necessarily cause the distances between encoded sites to grow with $n$. In a system of interacting qutrits (for which the proof of Proposition 10 still holds), this means that scaling interactions can be necessary to overcome Lieb-Robinson bounds and ensure that correlations can spread sufficiently fast through the enlarged system. The following simple geometric lemma provides some intuition for a quantitative lower bound on the growing length scales in such situations.

**Lemma 11.** Let $\{x_i\}_{i=1}^n$ be the points in a hypercube of side length $L \sim n^{1/D}$ in the square lattice $x_i \in \mathbb{Z}^D$. Let $\mathcal{E} : x_i \mapsto X_i \subseteq \mathbb{Z}^D$ be a map which encodes each point $x_i$ into a connected set of points in $\mathbb{Z}^D$ such that $|X_i| = \Omega(n^\alpha)$ and $X_i \cap X_j = \emptyset$. Let $d(x, y) : \mathbb{Z}^D \times \mathbb{Z}^D \to \mathbb{Z}$ be the taxicab metric on $\mathbb{Z}^D$.

For a radius $R = O(L)$, and any $y \in \mathbb{Z}^D$, the number of encoded points intersecting with the ball of radius $R$ centred at $y$ is bounded by

$$|B_R(y)| : = |\{X_i : \exists x \in X_i \text{ with } d(x,y) \leq R\}| = O\left(n^{1-\min\{\alpha, 1/D\}}\right). \tag{12}$$

Letting $\lambda = \min\{\alpha, 1/D\}$, we see that there are at most $O(n^{1-\lambda})$ sites $X_j$ within radius $R = O(L) = O(n^{1/D})$ of any $X_i$. On the other hand, there are at least $\Omega(n^{1-\lambda})$ of the $x_j$ within radius $O(n^{(1-\lambda)/D})$ of $x_i$ in the original lattice. In particular, this implies that there exist a pair of sites $x_i, x_j$ with $d(x_i, x_j) = O(n^{(1-\lambda)/D})$ whose encodings have $d(X_i, X_j) = \Omega(n^{1/D})$—in the encoded system, the distance is increased by a factor of $n^{\lambda/D}$.

The scalings here apply as a result of the requirement that analogue quantum simulators reproduce the dynamics of a target system. In other situations, such as adiabatic quantum simulation in which an approximately simulated ground state is the only requirement, encodings with superlinear qubit overhead are possible[53,54].

**Noisy analogue simulators**. Suppose we have quantum channels $T_t$, for $t \in [0, t_{\max}]$ which simulate some $H \in \mathrm{Herm}(\mathcal{H})$ with respect to $\Omega_{\mathrm{state}}$ and $\Omega_{\mathrm{obs}}$ up to accuracy $\epsilon$ as in Definition 5, corresponding to encoding maps $\mathcal{E}_{\mathrm{state}}$ and $\mathcal{E}_{\mathrm{obs}}$.

In practice, the experimental setup will suffer from noise in the steps of state preparation, evolution, and measurement. This will

correspond to noisy versions of the above maps, which we denote by $\tilde{T}_t$, $\tilde{\mathcal{E}}_{state}$, and $\tilde{\mathcal{E}}_{obs}$. For any $O \in \Omega_{obs}$, $\rho \in \Omega_{state}$, we may bound the additional error in observable expectation values incurred by the noisy maps by

$$
\begin{aligned}
&|\mathrm{tr}[\mathcal{E}_{obs}(O)(T_t \circ \mathcal{E}_{state})(\rho)] - \mathrm{tr}[\tilde{\mathcal{E}}_{obs}(O)(\tilde{T}_t \circ \tilde{\mathcal{E}}_{state}(\rho)]| \\
&= |\mathrm{tr}[O\left(\mathcal{E}^*_{obs} \circ T_t \circ \mathcal{E}_{state} - \tilde{\mathcal{E}}^*_{obs} \circ \tilde{T}_t \circ \tilde{\mathcal{E}}_{state}\right)(\rho)] \\
&\leq \| \mathcal{E}^*_{obs} \circ T_t \circ \mathcal{E}_{state} - \tilde{\mathcal{E}}^*_{obs} \circ \tilde{T}_t \circ \tilde{\mathcal{E}}_{state} \|_{1 \to 1} \\
&\leq \| \mathcal{E}^*_{obs} - \tilde{\mathcal{E}}^*_{obs} \|_{1 \to 1} + \| T_t - \tilde{T}_t \|_{1 \to 1} + \| \mathcal{E}_{state} - \tilde{\mathcal{E}}_{state} \|_{1 \to 1},
\end{aligned}
\tag{13}
$$

where $\|\cdot\|_{1 \to 1}$ denotes the one-to-one norm $\| \Lambda \|_{1 \to 1} = \sup_\rho \| \Lambda(\rho) \|_1$ (defined as the induced trace norm[55]—note that this is in particular upper bounded by the diamond norm), and $\mathcal{E}^*$ denotes the Hilbert-Schmidt dual of a superoperator $\mathcal{E}$. Hence the noisy simulator $\tilde{T}_t$ also simulates $H$ with respect to $\Omega_{state}$ and $\Omega_{obs}$, up to error

$$
\epsilon' \leq \epsilon + \sup_t \| T_t - \tilde{T}_t \|_{1 \to 1} + \| \mathcal{E}_{state} - \tilde{\mathcal{E}}_{state} \|_{1 \to 1} + \| \mathcal{E}^*_{obs} - \tilde{\mathcal{E}}^*_{obs} \|_{1 \to 1}.
\tag{14}
$$

**Local Hamiltonian simulation in a subspace.** Suppose that $H'$ is a $(\Delta, \eta, \epsilon)$-simulation of $H$ as defined by Cubitt et al.[11] (Definition 4), corresponding to encodings $\mathcal{E}_{state}$ and $\mathcal{E}_{obs}$, with the projector $Q = 0$. Here we show that the time evolution channel under $H'$, $(\cdot) \mapsto e^{-itH'}(\cdot)e^{itH'}$ gives a simulation in our sense, Definition 5.

We make use of the following lemmas. Lemma 12 ensures that measurement and time evolution are consistent with the encodings of Definition 4, and Lemma 13 bounds the error of $(\Delta, \eta, \epsilon)$-simulations under time evolution.

**Lemma 12.** (Cubitt et al., Proposition 4[11]) If $\mathcal{E}_{state}$ and $\mathcal{E}_{obs}$ are encodings as in Definition 4 and (2), then for all observables $O$ and states $\rho$ on the target system $\mathcal{H}$,

$$
\mathrm{tr}[\mathcal{E}_{obs}(O)\mathcal{E}_{state}(\rho)] = \mathrm{tr}[O\rho].
\tag{15}
$$

Moreover if the encoding is standard (rank$(P) > 0$ in Definition 4) then

$$
e^{-i\mathcal{E}_{obs}(H)t}\mathcal{E}_{state}(\rho)e^{i\mathcal{E}_{obs}(H)t} = \mathcal{E}_{state}\left(e^{-iHt}\rho e^{iHt}\right).
\tag{16}
$$

**Lemma 13.** (Cubitt et al., Proposition 28[11]) Let $H'$ be a $(\Delta, \eta, \epsilon)$-simulation of $H$ in the sense of Definition 4 corresponding to encodings $\mathcal{E}_{obs}$, $\mathcal{E}_{state}$. If $\rho'$ is a state in the simulator system $\mathcal{H}'$ satisfying $\mathcal{E}_{obs}(\mathrm{I})\rho' = \rho'$, then for all $t$

$$
\| e^{-iH't}\rho' e^{iH't} - e^{-i\mathcal{E}_{obs}(H)t}\rho' e^{i\mathcal{E}_{obs}(H)t} \|_1 \leq 2\epsilon t + 4\eta.
\tag{17}
$$

Combining these lemmas, we see that for any observable $O$ and state $\rho$ on $\mathcal{H}$,

$$
\begin{aligned}
&|\mathrm{tr}[\mathcal{E}_{obs}(O)e^{-iH't}\mathcal{E}_{state}(\rho)e^{iH't}] - \mathrm{tr}[Oe^{-iHt}\rho e^{iHt}]| \\
&= \left|\mathrm{tr}\left[\mathcal{E}_{obs}(O)\left(e^{-iH't}\mathcal{E}_{state}(\rho)e^{iH't} - e^{-i\mathcal{E}_{obs}(H)t}\mathcal{E}_{state}(\rho)e^{i\mathcal{E}_{obs}(H)t}\right)\right]\right| \\
&\leq \| O \| (2\epsilon t + 4\eta).
\end{aligned}
\tag{18}
$$

Hence the channels $T_t : \rho' \mapsto e^{-iH't}\rho' e^{iH't}$, for $t \in [0, t_{max}]$ simulate $H$ in the sense of Definition 5 with respect to any $\Omega_{state}$ and $\Omega_{obs}$, up to error

$$
\epsilon' \leq 2\epsilon t_{max} + 4\eta.
\tag{19}
$$

This provides some consistency between existing work and our notion of simulation; we have shown that evolution under a simulator Hamiltonian in the sense of Cubitt et al.[11] constitutes an analogue quantum simulator in our framework given by Definition 5.

**Short-time simulation with Lieb-Robinson bounds.** One advantage of only requiring the simulation of a particular set of observables $\Omega_{obs}$ in Definition 5, as opposed to reproducing the entire physical system, is that one can take advantage of the limited spread of correlations for short-time dynamics[24]. The idea of exploiting Lieb-Robinson bounds to reduce necessary hardware overhead has already been considered for the study of many-body quantum states on quantum computers[25,26], and more recently in the setting of analogue simulators[27]. We explain here how the latter fits into our framework.

Consider the case of a Hamiltonian $H_n$ on a $d$-dimensional lattice of $n$ qubits $\mathcal{H} \cong (\mathbb{C}^2)^{\otimes n}$, such that

$$
H_n = \sum_{x=1}^{n} h_x,
\tag{20}
$$

where the $h_x$ is a nearest-neighbour local interaction with $\|h_x\| \leq 1$, translated to position $x$ in the lattice, so that $H_n$ is translationally invariant.

If one is only interested in simulating the finite-time dynamics of a few local observables $\Omega_{obs}$ which are contained within a small neighbourhood of the origin, starting from a state $\rho = |0\rangle\langle 0|^{\otimes n}$, then it is sufficient (up to exponentially small error) to simulate a far smaller subsystem, corresponding to the Lieb-Robinson light cone, as in Fig. 7. This situation is studied by Trivedi et al.[27], in particular for the thermodynamic limit $n \to \infty$.

Let $H_m = \sum_{y=1}^{m} h_y$ be the simulator Hamiltonian, defined identically to $H_n$ but on a lattice of size $m < n$, $\mathcal{H}' \cong (\mathbb{C}^2)^{\otimes m}$. We encode $\rho$ and $O$ simply by restricting them to the smaller subsystem. Then a simulation

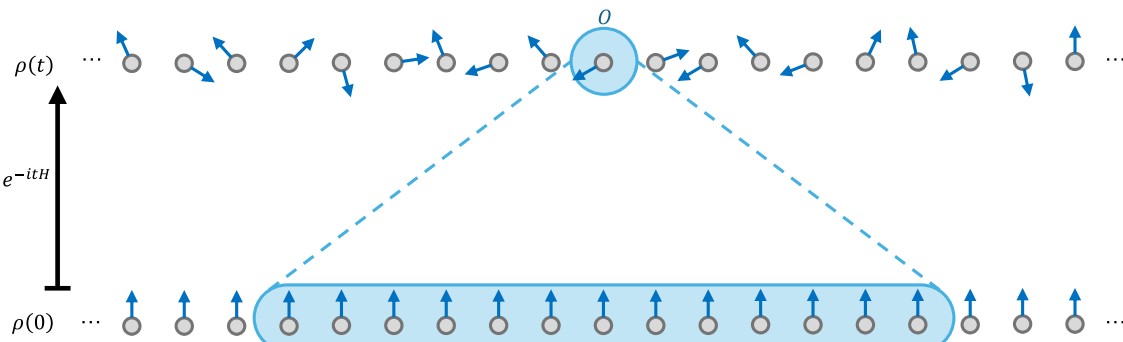

**Fig. 7 | Simulation with Lieb-Robinson bounds.** Simulation of a 1-dimensional spin system under a Hamiltonian $H$ for time $t$. In theory, the system extends infinitely, but to estimate the value of a local observable $O$ it is only necessary to simulate a subsystem corresponding to the Lieb-Robinson light cone.

of an observable $O \in \Omega_{\mathrm{obs}}$ up to accuracy $\epsilon$, satisfying

$$|\mathrm{tr}[Oe^{-iH_n t}\rho e^{iH_n t}] - \mathrm{tr}[\mathcal{E}_{\mathrm{obs}}(O)e^{-iH_m t}\mathcal{E}_{\mathrm{state}}(\rho)e^{iH_m t}]| \le \epsilon, \qquad (21)$$

can be accomplished in the large-$n$ regime for all $t \in [0,t_{\mathrm{max}}]$ if one takes $m = O\left(\log^d(1/\epsilon) + t_{\mathrm{max}}^d\right)$ (see Trivedi et al.[27], Lemma 1).

## Modular encodings and gadgets

In this section, we focus on the case of a simulator channel $T_t$ given by time evolution under a local simulator Hamiltonian $H'$, which should reproduce the dynamics of the local target Hamiltonian $H = \sum_i H_i$. In light of the size-independence requirement of Definition 1., it is natural to encode each $H_i$ term separately into some term $H'_i$, but systematically doing so is a non-trivial task: we need the encoded terms to interact with each other in a way that mimics the original system.

This problem can be tackled using perturbative gadgets. Perturbative gadgets were initially introduced by Kempe et al.[42] as a means of proving QMA-completeness of the 2-local Hamiltonian problem by reduction from the three-local case[30], and have since been used extensively in the field of Hamiltonian complexity theory. In this work, we especially focus on the use of gadgets for Hamiltonian locality reduction, though it should be noted that perturbative gadgets can also be used to simplify the structure of the interaction hypergraph[17] and, in general, to reduce Hamiltonians to more restrictive families of interactions[44,56,57]. Moreover, beyond Hamiltonian complexity-theoretic results, gadgets can be tailored to improve the performance of variational quantum algorithms[34].

In this work, we introduce a formalism which we argue encompasses any attempt at gadgetisation, in a sense which we make precise Definition (14), in order to prove general properties of such constructions. Note that our approach, and the $(\eta, \epsilon)$ accuracy parameters, are closely related to those used in other definitions of simulation[11,18]. We refine the approach of the latter by generalising to a potentially non-perturbative regime and by considering the feature of combining well with other interactions as a generic requirement for gadgets. We use these results to argue that any size-independent encoding of a Hamiltonian $H$ into another $H'$ cannot reduce the locality of interactions (for example, reducing a 3-local Hamiltonian to a 2-local Hamiltonian).

The setup is as follows: we consider a large system $\mathcal{H} = \otimes_{i=1}^n \mathcal{H}_i$, within which a local interaction $H \in \mathrm{Herm}(\mathcal{H})$ acts on a subsystem of $O(1)$ sites. With the introduction of a small ancillary system $\mathcal{A}$, we aim to replace $H$ by some gadget $H' \in \mathrm{Herm}(\mathcal{H} \otimes \mathcal{A})$, which acts on $O(1)$ sites in $\mathcal{H}$ and $\mathcal{A}$.

A simulator Hamiltonian in the sense of Definition 5 need not necessarily capture the entire spectrum of its target Hamiltonian. In this case, however, we are thinking of $H$ as a single interaction in a larger system, and as such we cannot generally assume that its eigenspaces will be preserved under time evolution. Therefore, we require as a minimum that $H'$ should (when restricted to some subspace defined by a projector $P'$) approximately reproduce the full spectrum of $H$. Moreover, for $H'$ to be a useful gadget, it must combine well with other Hamiltonian terms acting on $\mathcal{H}$. That is to say, there should exist $P' \in \mathrm{Proj}(\mathcal{H} \otimes \mathcal{A})$ such that $P'(H' + H_{\mathrm{else}} \otimes \mathrm{I})P'$ approximates the spectrum of $H + H_{\mathrm{else}}$, for any $H_{\mathrm{else}} \in \mathrm{Herm}(\mathcal{H})$ (see Fig. 3a). We formalise this with the following definition.

**Definition 14.** (($\zeta, \epsilon$)-gadget property) Given a Hamiltonian $H \in \mathrm{Herm}(\mathcal{H})$ acting on a system $\mathcal{H} = \otimes_{i=1}^n \mathcal{H}_i$, and $H' \in \mathrm{Herm}(\mathcal{H} \otimes \mathcal{A})$ for $\mathcal{A}$ an ancillary system, we say that $(H', \mathcal{A})$ satisfies the ($\zeta, \epsilon$)-gadget property for $H$ if there exists $P' \in \mathrm{Proj}(\mathcal{H} \otimes \mathcal{A})$, $\tilde{P} \in \mathrm{Proj}(\mathcal{A}) \setminus \{0\}$ such that, for any $H_{\mathrm{else}} \in \mathrm{Herm}(\mathcal{H})$, there exists a unitary $\tilde{U}_{H_{\mathrm{else}}} \in \mathrm{U}(\mathcal{H} \otimes \mathcal{A})$

with

$$\| P'(H' + H_{\mathrm{else}} \otimes \mathrm{I})P' - \tilde{U}_{H_{\mathrm{else}}}\left((H + H_{\mathrm{else}}) \otimes \tilde{P}\right)\tilde{U}_{H_{\mathrm{else}}}^\dagger \| \le \epsilon + \zeta \| H_{\mathrm{else}} \| . \qquad (22)$$

In other words, $(H', \mathcal{A})$ satisfies the ($\zeta, \epsilon$)-gadget property for $H$ if, when restricted a subspace defined by $P'$, $H' + H_{\mathrm{else}} \otimes \mathrm{I}$ approximates the spectrum of $H + H_{\mathrm{else}}$ up to error $\epsilon + \zeta\|H_{\mathrm{else}}\|$. Notice that $\tilde{P}$ is almost arbitrary; its rank determines the multiplicity of each eigenvalue of $H + H_{\mathrm{else}}$ in the simulator system, but otherwise it can be rotated by $\tilde{U}_{H_{\mathrm{else}}}$, which rotates the eigenvectors of $(H + H_{\mathrm{else}}) \otimes \tilde{P}$ approximately onto those of $P'(H' + H_{\mathrm{else}} \otimes \mathrm{I})P'$.

As noted by Cubitt et al.[11], there are two distinct types of gadgets used in literature:

- Mediator gadgets, in which ancillary qubits are inserted between logical qubits to mediate interactions, and
- Subspace gadgets, in which single logical qubits are encoded into several physical qubits, restricted to a two-dimensional subspace by strong interactions.

Definition 14 encompasses the former, but not the latter. Qualitatively this is because whereas mediator gadgets replace interactions, subspace gadgets replace entire qubits, including all of the interactions they take part in. It would be possible to extend our formalism to subspace gadgets, by restricting the range of $H_{\mathrm{else}}$ in Definition 14 to terms, which do not interact with the target qubit. We do not consider this here, however, for brevity and because subspace gadgets do not reduce the locality of interactions, which is our primary motivation for this section.

Although Definition 14 is a natural requirement, it is not convenient to work with due to the appearance of the general $H_{\mathrm{else}}$ acting on the entire of $\mathcal{H}$, upon which $\tilde{U}$ depends. The following alternative definition does not suffer from this problem.

**Definition 15.** (($\eta, \epsilon$)-gadget) Let $H \in \mathrm{Herm}(\mathcal{H})$ be a Hamiltonian on a Hilbert space $\mathcal{H}$, and let $\mathcal{A}$ be an ancillary Hilbert space. For $H' \in \mathrm{Herm}(\mathcal{H} \otimes \mathcal{A})$, we say that $(H', \mathcal{A})$ is a ($\eta, \epsilon$)-gadget for $H$ if there exists $P \in \mathrm{Proj}(\mathcal{A}) \setminus \{0\}$ and $U \in \mathrm{U}(\mathcal{H} \otimes \mathcal{A})$ such that

$$\| U - \mathrm{I} \| \le \eta, \quad \| P'H'P' - U(H \otimes P)U^\dagger \| \le \epsilon, \qquad (23)$$

where $P' = U(\mathrm{I} \otimes P)U^\dagger \in \mathrm{Proj}(\mathcal{H} \otimes \mathcal{A})$.

The advantage of Definition 15 is that it is stated in terms of a local rather than global property. Assuming that $H, H', P'$ act on only $O(1)$ sites in $\mathcal{H}$ and $\mathcal{A}$, we can without loss of generality restrict to this significantly smaller subspace to check whether $H'$ is a gadget. This is in contrast with Definition 14, which requires us to in principle consider interactions over the full $n$-site space in order to check the gadget property.

To motivate the use of Definition 15, we show that the above notions are in correspondence; things that look like gadgets are always gadgets, and vice-versa. This is formalised by the following two theorems, proved in Supplementary Note 3.

**Theorem 16.** (($\eta, \epsilon$)-gadgets have the ($\zeta, \epsilon$)-gadget property) Suppose that $(H', \mathcal{A})$ is a ($\eta, \epsilon$)-gadget for $H$. Then $(H', \mathcal{A})$ satisfies the ($\zeta, \epsilon$)-gadget property for $H$, where $\zeta = O(\eta)$.

**Theorem 17.** (The ($\zeta, \epsilon$)-gadget property requires a ($\eta, \epsilon$)-gadget) Suppose that $(H', \mathcal{A})$ satisfies the ($\zeta, \epsilon$)-gadget property for $H$, where $H$, $H'$, and $P'$ act on $O(1)$ sites in $\mathcal{H} = \otimes_{i=1}^n \mathcal{H}_i$. Then $(H', \mathcal{A})$ is a ($\eta, \epsilon'$)-gadget for $H$, where $\eta = O(\epsilon) + O(\zeta^{\frac{1}{2}})$ and $\epsilon' = O(\epsilon) + O(\zeta)$.

The roles of the $\eta$ and $\epsilon$ parameters are to bound the error in the eigenvectors and eigenvalues respectively. Roughly speaking, $\eta$

quantifies how well the gadget combines with other terms, and $\epsilon$ quantifies the spectral error of the gadget in isolation. A good gadget requires both of these parameters to be small. In the next section we present a 3-to-2 local gadget which is an extreme case of this, with $\epsilon = 0$ at the cost of a large $\eta$ error.

Prior work in Hamiltonian complexity theory has focused on gadgetisation in the context of ground state estimation[18,30,44] or simulation in a low-energy subspace[11]; as a result, a case of particular relevance is when $P'$ projects onto the low-energy subspace of $H'$. For $\Delta \in \mathbb{R}$, we write $P_{\leq \Delta(H')}$ for the projector onto the span of the eigenvectors of $H'$ with eigenvalues in the range $(-\infty, \Delta]$.

**Definition 18.** (($\Delta, \eta, \epsilon$)-gadget) Let $H \in \text{Herm}(\mathcal{H})$ be a Hamiltonian on a Hilbert space $\mathcal{H}$, and let $\mathcal{A}$ be an ancillary Hilbert space. For $H' \in \text{Herm}(\mathcal{H} \otimes \mathcal{A})$, we say that $(H', \mathcal{A})$ is a $(\Delta, \eta, \epsilon)$-gadget for $H$ if there exists $P \in \text{Proj}(\mathcal{A}) \setminus \{0\}$, and $U \in \text{U}(\mathcal{H} \otimes \mathcal{A})$ such that $P_{\leq \Delta(H')} = U(\text{I} \otimes P)U^\dagger$, and

$$\| U - \text{I} \| \leq \eta, \quad \| P_{\leq \Delta(H')} H' P_{\leq \Delta(H')} - U(H \otimes P)U^\dagger \| \leq \epsilon. \quad (24)$$

In other words, the pair $(H', \mathcal{A})$ satisfy Definition 15, in the special case where we can use $P' = P_{\leq \Delta(H')}$.

Notice that Definition 18 imposes a significantly stronger requirement on $H'$ than Definition 15; a priori there is no reason to expect that there will exist any choice of $P$ and $U$ such that $P_{\leq \Delta(H')} = U(\text{I} \otimes P)U^\dagger$. Definitions Definition 15 and Definition 18 are sufficient to guarantee desirable combination properties, and are satisfied by widely-used constructions.

### Examples of gadgets

Lemmas 4–7 of Bravyi et al.[18] can be naturally adapted to give several constructions for $(\Delta, \eta, \epsilon)$ gadgets, which we use to demonstrate that Definition 15 encompasses commonly-used techniques. In the following we take $\mathcal{H}' = \mathcal{H} \otimes \mathcal{A}$, and $\mathcal{A} \cong \mathbb{C}^2$. For $V$ an operator on $\mathcal{H}'$ we write it in block-diagonal form with respect to the basis of $\mathcal{A}$ as

$$V = \begin{pmatrix} V_{00} & V_{01} \\ V_{10} & V_{11} \end{pmatrix}, \quad (25)$$

where, for instance, $V_{00} = (\text{I} \otimes \langle 0|)V(\text{I} \otimes |0\rangle)$.

**Lemma 19.** (First-order gadgets, adapted from Bravyi et al.[18]) Suppose $H \in \text{Herm}(\mathcal{H})$ and $V \in \text{Herm}(\mathcal{H}')$ are such that

$$\| H - V_{00} \| \leq \frac{\epsilon}{2}. \quad (26)$$

Then $H' = \Delta H_0 + V$ defines a $(O(\Delta), \eta, \epsilon)$-gadget for $H$, where $H_0 = \text{I} \otimes |1\rangle\langle 1|$, provided that $\Delta \geq O(\epsilon^{-1}\|V\|^2 + \eta^{-1}\|V\|)$.

**Lemma 20.** (Second-order gadgets, adapted from Bravyi et al.[18]) Let $H \in \text{Herm}(\mathcal{H})$, and suppose $V^{(1)}, V^{(0)} \in \text{Herm}(\mathcal{H}')$ are such that $\|V^{(1)}\|, \|V^{(0)}\| \leq \Lambda$, $V_{10}^{(0)} = V_{01}^{(0)} = V_{00}^{(1)} = 0$, and

$$\| H - V_{00}^{(0)} + V_{01}^{(1)} V_{10}^{(1)} \| \leq \frac{\epsilon}{2}. \quad (27)$$

Then $H' = \Delta H_0 + \Delta^{\frac{1}{2}} V^{(1)} + V^{(0)}$ is a $(O(\Delta), \eta, \epsilon)$-gadget for $H$, where $H_0 = \text{I} \otimes |1\rangle\langle 1|$, if

$$\Delta \geq O(\epsilon^{-2}\Lambda^6 + \eta^{-2}\Lambda^2). \quad (28)$$

**Lemma 21.** (Third-order gadgets, adapted from Bravyi et al.[18]) Let $H \in \text{Herm}(\mathcal{H})$, and suppose $V^{(2)}, V^{(1)}, V^{(0)} \in \text{Herm}(\mathcal{H}')$ are such that $\|V^{(2)}\|, \|V^{(1)}\|, \|V^{(0)}\| \leq \Lambda$, $V_{10}^{(1)} = V_{01}^{(1)} = V_{10}^{(0)} = V_{01}^{(0)} = 0$, $V_{00}^{(2)} = 0$,

$$\| H - V_{00}^{(0)} - V_{01}^{(2)} V_{11}^{(2)} V_{10}^{(2)} \| \leq \frac{\epsilon}{2}, \quad \text{and} \quad V_{00}^{(1)} = V_{01}^{(2)} V_{10}^{(2)}. \quad (29)$$

Then $H' = \Delta H_0 + \Delta^{\frac{2}{3}} V^{(2)} + \Delta^{\frac{1}{3}} V^{(1)} + V^{(0)}$ is a $(O(\Delta), \eta, \epsilon)$-gadget for $H$, where $H_0 = \text{I} \otimes |1\rangle\langle 1|$, if

$$\Delta \geq O(\epsilon^{-3}\Lambda^{12} + \eta^{-3}\Lambda^3). \quad (30)$$

We illustrate the application of these lemmas to our definition with the following ubiquitous gadgets from Oliviera et al.[17]:

Given a target Hamiltonian $H = A \otimes B \in \text{Herm}(\mathcal{H}_A \otimes \mathcal{H}_B)$, the subdivision gadget on $\mathcal{H}_A \otimes \mathcal{H}_B \otimes \mathcal{H}_C$ (where $\mathcal{H}_C \cong \mathbb{C}^2$) is defined by

$$H' = \Delta H_0 + \Delta^{\frac{1}{2}} V^{(1)} + V^{(0)}, \quad (31)$$

where

$$H_0 = \text{I} \otimes \text{I} \otimes |1\rangle\langle 1|, \quad (32)$$

$$V^{(1)} = \frac{1}{\sqrt{2}}(-A \otimes \text{I} + \text{I} \otimes B) \otimes X, \quad (33)$$

$$V^{(0)} = \frac{1}{2}(A^2 \otimes \text{I} + \text{I} \otimes B^2) \otimes \text{I}. \quad (34)$$

Then by Lemma 20 we see that, for sufficiently large $\Delta$, $(H', \mathcal{H}_C)$ defines a $(O(\Delta), \eta, \epsilon)$-gadget for $H$ (see Fig. 8a).

Given a target Hamiltonian $H = A \otimes B \otimes C \in \text{Herm}(\mathcal{H}_A \otimes \mathcal{H}_B \otimes \mathcal{H}_C)$, the 3-to-2 local gadget on $\mathcal{H}_A \otimes \mathcal{H}_B \otimes \mathcal{H}_C \otimes \mathcal{H}_D$ (where $\mathcal{H}_D \cong \mathbb{C}^2$) is defined by

$$H' = \Delta H_0 + \Delta^{\frac{2}{3}} V^{(2)} + \Delta^{\frac{1}{3}} V^{(1)} + V^{(0)}, \quad (35)$$

where

$$H_0 = \text{I} \otimes \text{I} \otimes \text{I} \otimes |1\rangle\langle 1|, \quad (36)$$

$$V^{(2)} = \frac{1}{\sqrt{2}}(-A \otimes \text{I} + \text{I} \otimes B) \otimes \text{I} \otimes X - \text{I} \otimes \text{I} \otimes C \otimes |1\rangle\langle 1|, \quad (37)$$

$$V^{(1)} = \frac{1}{2}(-A \otimes \text{I} + \text{I} \otimes B)^2 \otimes \text{I} \otimes \text{I}, \quad (38)$$

$$V^{(0)} = \frac{1}{2}(A^2 \otimes \text{I} + \text{I} \otimes B^2) \otimes C \otimes \text{I}. \quad (39)$$

By Lemma 20 we see that, for sufficiently large $\Delta$, $(H', \mathcal{H}_D)$ defines a $(O(\Delta), \eta, \epsilon)$-gadget for $H$ (see Fig. 8b).

We provide the following example to illustrate the importance of the $\eta$ parameter as a quantifier of how well a gadget combines with other terms.

Let $H = A \otimes B \otimes C \in \text{Herm}((\mathbb{C}^2)^{\otimes 3})$ be a 3-qubit interaction, and diagonalise $A$, $B$, and $C$ as

$$A = \lambda_0^A |0\rangle\langle 0| + \lambda_1^A |1\rangle\langle 1|, \quad B = \lambda_0^B |0\rangle\langle 0| + \lambda_1^B |1\rangle\langle 1|, \quad C = \lambda_0^C |0\rangle\langle 0| + \lambda_1^C |1\rangle\langle 1|. \quad (40)$$

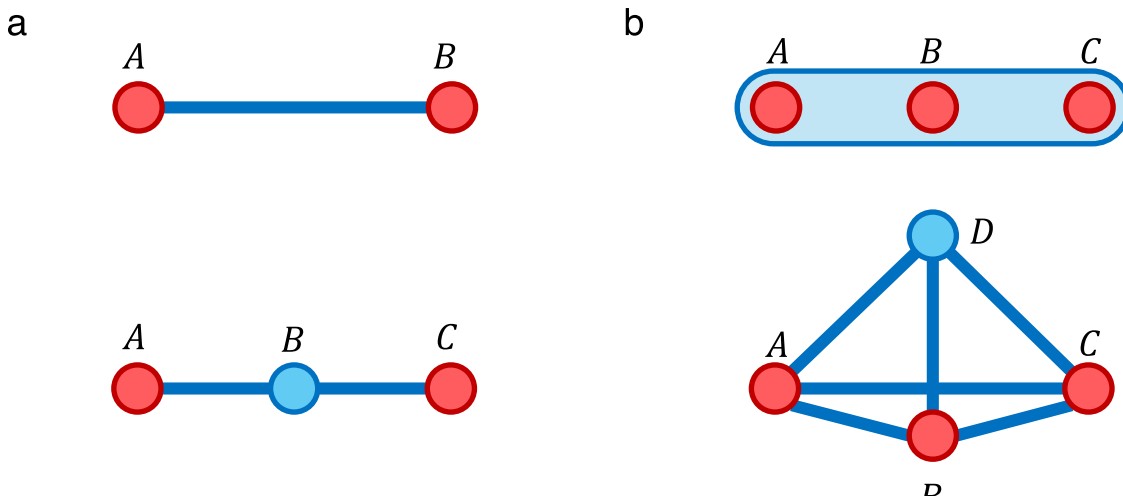

**Fig. 8 | Existing gadgets. a** Interaction hypergraphs of a 2-system interaction before (above) and after (below) the use of the subdivision gadget. **b** Interaction hypergraphs of a 3-system interaction before (above) and after (below) the use of the 3-to-2 gadget.

Let $H' \in \mathrm{Herm}((\mathbb{C}^2)^{\otimes 4})$ be defined as

$$
\begin{aligned}
H' = &\; \lambda_0^B (A - \lambda_0^A \mathrm{I}) \otimes \mathrm{I} \otimes \mathrm{I} \otimes C \\
&+ \lambda_1^B \mathrm{I} \otimes (A - \lambda_0^A \mathrm{I}) \otimes \mathrm{I} \otimes C \\
&+ \lambda_0^A \mathrm{I} \otimes \mathrm{I} \otimes B \otimes C \, ,
\end{aligned}
\tag{41}
$$

and let $P' \in \mathrm{Proj}((\mathbb{C}^2)^{\otimes 4})$ be

$$
P' = (\mathrm{I} \otimes |0\rangle\langle 0| \otimes |0\rangle\langle 0| + |0\rangle\langle 0| \otimes \mathrm{I} \otimes |1\rangle\langle 1|) \otimes \mathrm{I} .
\tag{42}
$$

Then in fact the restriction of $H'$ to the image of $P'$ exactly reproduces the spectrum of $H$. This hence defines a 3-to-2 $(\eta, 0)$-gadget —or a $(\Delta, \eta, 0)$-gadget, if one adds a term of the form $O(\Delta)(\mathrm{I} - P')$ to $H'$. The caveat is that this gadget has a large $\eta$ parameter, and hence it does not combine well with other interactions. For instance, in Definition 15 we might take $P = |0\rangle\langle 0| \otimes \mathrm{I} \otimes \mathrm{I} \otimes \mathrm{I}$, and $U = (\mathbb{F} \otimes |0\rangle\langle 0| + \mathrm{I} \otimes \mathrm{I} \otimes |1\rangle\langle 1|) \otimes \mathrm{I}$, where $\mathbb{F}$ is the two-qubit swapping operator. This gives $\eta = 2$.

The construction of $H'$ can be thought of as splitting the $A$ qubit into two qubits (see Fig. 9), and controlling whether the first or second qubit is excited depending on the value of the $B$ qubit. Therefore, if the full Hamiltonian contains another interaction term that acts on the $A$ site in $H$, then the locality of this term will be increased under the gadgetisation procedure. Such a gadget cannot be used to systematically reduce the locality of a Hamiltonian with many interactions.

### Gadget combination results

The following results show that gadgets satisfying Definition 15 or Definition 18 can be systematically combined as desired. Our techniques and proofs extend prior work[17,31,58], using the convenient formalism of the direct rotation[59]. The scalings of the parameters $\eta', \epsilon'$ are not necessarily optimal, though they sufficient for application to the subdivision and 3-to-2 gadget constructions exhibited above. The proofs of our gadget combination results can be found in Supplementary Note 4.

We summarise the setup below, which will be used throughout the following results.

**Setup 22.** Let $H \in \mathrm{Herm}(\mathcal{H})$ be a Hamiltonian on $n$ sites, $\mathcal{H} = \otimes_{i=1}^n \mathcal{H}_i$. Assume $H = \sum_{i=1}^N H_i$, where $N = O(n)$, such that each $H_i$ acts on at most $k = O(1)$ of the sites $\mathcal{H}_i$, and each site participates in at most $d = O(1)$

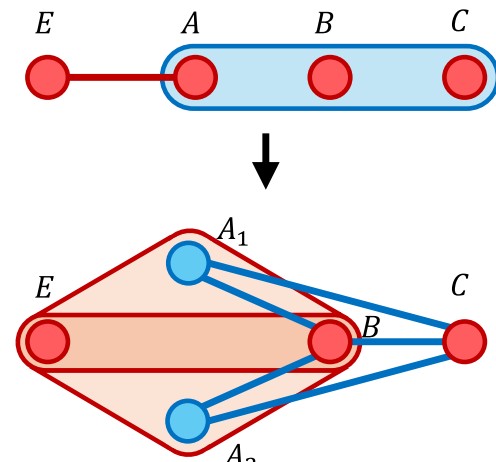

**Fig. 9 | The exact 3-to-2 gadget.** The (blue) 3-local interaction between $A$, $B$, and $C$ is replaced by a series of (blue) 2-local interactions, where the $A$ site has been split into two sites $A_1$ and $A_2$. However, after this process, the 2-local interaction (red) between $A$ and another qubit $E$ is replaced by two 3-local interactions between $E, A_1, B$ and $E, A_2, B$. Compare this with Fig. 8b, for which additional interactions on qubit $A$ will remain on qubit $A$ of the gadgetised Hamiltonian without any need for adjustment.

interactions. Assume also that $H$ has bounded interaction strengths, that is, $\|H_i\| \le J$ for all $i$.

In the below propositions we consider a family (depending on $n$) of gadgets $(H_i', \mathcal{A}_i)$ for $H_i$, with $U_i$, $P_i$, and $P_i'$ defined as in Definition 15, for each $i$. Assume that $\mathcal{A}_i$ consists of $O(1)$ ancillary sites and that $H_i'$ is a local Hamiltonian consisting of $O(1)$ interactions, such that

$$
\| H_i' \| \le J' , \quad \| (\mathrm{I} \otimes P_i) H_i' (\mathrm{I} \otimes P_i^\perp) \| \le J_O' .
\tag{43}
$$

Firstly, we state the main result: that gadgets as in Definition 15 may be systematically combined to produce new gadgets.

**Proposition 23.** (Parallel $(\eta, \epsilon)$-gadget combination) Let $H = \sum_i H_i$ be as in Setup 22, and suppose that each $(H_i', \mathcal{A}_i)$ defines a $(\eta, \epsilon)$-gadget for $H_i$.

Define

$$H' = \sum_i H'_i \in \mathrm{Herm}\left(\mathcal{H} \otimes (\otimes_i \mathcal{A}_i)\right). \tag{44}$$

Then $(H', \otimes_i \mathcal{A}_i)$ is a $(\eta', \epsilon')$-gadget for $H$, where

$$\epsilon' = O(n\epsilon + n\eta J + n\eta^3 J'_O + n\eta^4 J'), \quad \eta' = O(n\eta). \tag{45}$$

For completeness, we also prove a similar result that $(\Delta, \eta, \epsilon)$-gadgets can be combined to create a new $(\Delta', \eta', \epsilon')$-gadget. It follows from Proposition 23 that the combination of many $(\Delta, \eta, \epsilon)$-gadgets defines a $(\eta', \epsilon')$-gadget, however it still remains to show that the projector $P'$ in the sense of Definition 15 may be taken as a low-energy projector $P_{\leq \Delta'(H')}$.

**Proposition 24.** (Parallel $(\Delta, \eta, \epsilon)$-gadget combination) Let $H = \sum_i H_i$ be as in Setup 22,, and suppose that each $(H'_i, \mathcal{A}_i)$ defines a $(\Delta, \eta, \epsilon)$-gadget for $H_i$, where

$$\Delta \geq \frac{\|H\| + J + N(\epsilon + 2J\eta)}{\frac{1}{4} - 2\eta} = O(nJ), \tag{46}$$

and assume that the scaling of $\eta$ with $n$ is bounded as

$$\eta = o(n^{-\frac{1}{2}}), \tag{47}$$

and moreover that, for large $J'$,

$$n\epsilon + n\eta J + n\eta^3 J'_O + n\eta^4 J' = o(J'), \quad J' = O(\Delta). \tag{48}$$

Define

$$H' = \sum_i H'_i \in \mathrm{Herm}\left(\mathcal{H} \otimes (\otimes_i \mathcal{A}_i)\right). \tag{49}$$

Then $(H', \otimes_i \mathcal{A}_i)$ is a $(\Delta', \eta', \epsilon')$-gadget for $H$, where

$$\Delta' = \frac{1}{2}\Delta, \quad \epsilon' = O(n\epsilon + n\eta J + n\eta^3 J'_O + n^3\eta^4 J'), \quad \eta' = O(n\eta). \tag{50}$$

For an example of how these conditions can be satisfied, consider the case of combining many of the 3-to-2 gadgets described above. Setting $J = 1$ for convenience, we have $J' = \Theta(\Delta)$, $J'_O = \Theta(\Delta^{2/3})$, and $\epsilon, \eta = O(\Delta^{-1/3})$. The errors $\epsilon'$ and $\eta'$ both grow as $O(n\Delta^{-1/3})$, so a good gadget will require $\Delta = \Omega(n^3)$. A direct computation verifies that this condition also ensures that $((46)$–$(48))$ are satisfied. Hence reduction from a 3-local to 2-local Hamiltonian in this way requires interaction strengths to scale as $n^3$.

To combine $(\Delta, \eta, \epsilon)$ gadgets using Proposition 24 requires the unappealing conditions of $(46)$–$(47)$, which explicitly require the gadget energies to scale with $n$. In fact, as noted by Bravyi et al.[31], the regime of bounded-strength interactions does still allow approximation of the ground state energy of $H$—the caveat being that the errors are extensive. Below is a generalisation of their main result.

**Theorem 25.** (Ground state energy estimation with $(\Delta, \eta, \epsilon)$-gadgets, generalising Bravyi et al., Theorem 1[31]) Let $H = \sum_i H_i$ be as in Setup 22, and suppose that each $(H'_i, \mathcal{A}_i)$ defines a $(\Delta, \eta, \epsilon)$-gadget for $H_i$.
Define

$$H' = \sum_i H'_i \in \mathrm{Herm}\left(\mathcal{H} \otimes (\otimes_i \mathcal{A}_i)\right). \tag{51}$$

Then the ground state energies of $H$ and $H'$ satisfy

$$|\lambda_0(H) - \lambda_0(H')| = O(n\epsilon + n\eta J + n\eta^3 J'_O + n\eta^4 J'). \tag{52}$$

## Gadget energy scaling

Here we present the main result of the section: general locality reduction gadgets cannot exist without unfavourably scaling energies. This result holds in the most general setting of $(\eta, \epsilon)$-gadgets (Definition 15), and hence follows even from the relaxed $(\zeta, \epsilon)$-gadget property of Definition 14.

**Theorem 26.** (Gadget energy scaling) Let $\mathcal{H} = (\mathbb{C}^2)^{\otimes k}$ be the space of $k = O(1)$ qubits, and let $H$ be the $k$-fold tensor product of Pauli $Z$ operators with strength $J > 0$,

$$H = J \bigotimes_{i=1}^{k} Z_i. \tag{53}$$

Suppose $(H', \mathcal{A})$ is a $(\eta, \epsilon)$-gadget for $H$ for $H'$ a $k'$-local Hamiltonian, where $k' < k$.

Then, provided $\epsilon < J$, the gadget must have energy scale $\|H'\| \geq \frac{J-\epsilon}{\eta} = \Omega(\eta^{-1})$.

The method of proof (found in Supplementary Note 5) is simple, and very likely does not provide an optimal lower bound for $\|H'\|$, due to the lack of any dependence on $k$. We expect that such dependence should be present; any approach which iteratively lowers the locality of an interaction from $k$-local to 2-local will accumulate scalings from each round of gadgetisation, but this does not rule out a more direct approach. Existing methods to reduce locality, such as the subdivision and 3-to-2 gadgets of Oliviera et al.[17] and higher-order gadgets[34,60], give scalings that suggest that any $k$-to-2-local gadget construction should require energies which scale exponentially in $k$. The question of whether such exponential scaling is the best possible was first raised by Bravyi et al.[31], and is still unresolved. Using the formalism introduced here, this problem can be precisely stated, and optimisation of Theorem 26 may provide a negative result. Furthermore, we expect that it may be possible to answer similar questions about gadget energy scaling in other cases, for example in simplifying the structure of an interaction graph or reducing to smaller families of interactions.

The significance of Theorem 26 is that it essentially rules out a size-independent (Definition 1.) simulation of a $k$-local Hamiltonian $H$ by another $k'$-local Hamiltonian $H'$ for $k' < k$, for the following reason. Any modular encodings require the use of term-by-term gadgets, which must each satisfy the $(\zeta, \epsilon)$-gadget property (Definition 14) with $\zeta, \eta = O(n^{-1})$ to guarantee that they can be combined (since the rest of the Hamiltonian will have $\|H_{\mathrm{else}}\| = O(n)$). By Theorem 17, this requires the use of $(\eta, \epsilon)$-gadgets (Definition 15) with $\eta = O(n^{-1/2})$, and by Theorem 26 this will require interactions which scale at least as $\Omega(n^{1/2})$.

A couple of notes on gadget energy scalings in existing work: Bausch[61] gives a method to reduce the exponential or doubly-exponential scaling in perturbative Hamiltonians to polynomial scaling, and Cao et al.[33] present gadgets whose interaction strengths do not grow with accuracy. However, both cases violate size-independence (Definition 1.) in other ways such as polynomial scaling in the number of simulator qubits or instead shrinking the interaction strengths.

## Gadgets from the quantum Zeno effect

In this section, we demonstrate an alternative approach for reducing the locality of an interaction in a Hamiltonian—a task for which Theorem 26 establishes the need for energies which scale with the size of the system, when conventional gadgets are used. The construction presented here, however, uses the freedom afforded by the general

simulation channel $T_t$ in Definition 5 to take advantage of an additional resource: dissipation.

We will see that, despite some impractical features for experimental implementation, this approach offers a theoretical improvement in scalings over the conventional gadget techniques discussed earlier in the section. Additionally, this construction captures a key feature of our framework for analogue simulators given in Definition 5 in contrast with existing work: we define simulators in terms of their dynamic behaviour, rather than in terms of the properties of static Hamiltonians.

For the process we describe here, we repeatedly refer to measurement for conceptual simplicity when talking about probabilities, but this terminology is somewhat misleading; we do not record or use the outcome.

Let $H \in \mathrm{Herm}(\mathcal{H})$ be a single interaction in a many-body system, which we intend to simulate. As before, we will introduce an ancillary qubit $\mathcal{A} \cong \mathbb{C}^2$, and evolve under a Hamiltonian $H' \in \mathrm{Herm}(\mathcal{H} \otimes \mathcal{A})$, but now we supplement the natural time evolution with regular projective measurements on the $\mathcal{A}$ system at time intervals of $\delta t$. By the quantum Zeno effect[62], this forces the $\mathcal{A}$ system to stay in the $|0\rangle$ state with high probability, meanwhile simulating the desired interaction on the $\mathcal{H}$ system.

The following result, Proposition 27, provides a formal construction for the measurement-based gadgets described above—see Supplementary Note 6 for the proof. Qualitatively, this result tells us that if we evolve $|\psi\rangle \otimes |0\rangle$ for time $\delta t$ under the simulator Hamiltonian $H'$, and then measure the ancillary qubit, we will obtain a '1' result with probability $O((\delta t)^3)$ (corresponding to an amplitude of $O((\delta t)^{3/2})$). In the more likely case that we obtain '0', the post-measurement state (on the $\mathcal{H}$ space) is $e^{-i\delta t H}|\psi\rangle$, for some new Hamiltonian $H$, up to error $O((\delta t)^2)$. By repeating this process $t/\delta t$ times, we will hence obtain a state $e^{-itH}|\psi\rangle + O(t(\delta t))$ on the $\mathcal{H}$ space if '0' is measured in every round of measurement. The probability of a measurement error in this process scales as $t(\delta t)^2$, hence can be controlled provided that $\delta t = O(t^{-1/2})$, which will always be satisfied if we choose $\delta t = O(t^{-1})$ in order to control the error on the post-measurement state.

**Proposition 27.** For a Hilbert space $\mathcal{H}$ and an ancillary qubit $\mathcal{A} = \mathbb{C}^2$, let $H' \in \mathrm{Herm}(\mathcal{H} \otimes \mathcal{A})$ be a Hamiltonian given by

$$H' = H_{\mathrm{I}} \otimes \mathrm{I} + H_X \otimes X + H_{|1\rangle\langle 1|} \otimes |1\rangle\langle 1|, \tag{54}$$

for some $H_{\mathrm{I}}, H_X, H_{|1\rangle\langle 1|} \in \mathrm{Herm}(\mathcal{H})$ depending on a small parameter $\delta t$ such that $\|H_{\mathrm{I}}\| = O(1)$, $\|H_X\| = O((\delta t)^{-1/2})$, and $\| H_{|1\rangle\langle 1|} \| = O((\delta t)^{-1})$ with $H_{|1\rangle\langle 1|}^2 = \omega^2 \mathrm{I}$, $\omega = \frac{2\pi}{\delta t}$.

Then, for any $|\psi\rangle \in \mathcal{H}$,

$$e^{-i\delta t H'}(|\psi\rangle \otimes |0\rangle) = \left(e^{-i\delta t H}|\psi\rangle + O((\delta t)^2)\right) \otimes |0\rangle + O((\delta t)^{3/2}) \otimes |1\rangle, \tag{55}$$

where

$$H = H_{\mathrm{I}} - \omega^{-2} H_X H_{|1\rangle\langle 1|} H_X. \tag{56}$$

This provides a new 3-to-2-local gadget for Pauli strings. For example, we can set $H_{\mathrm{I}} = -Z_1$, $H_X = \sqrt{\frac{\omega}{2}}(Z_2 + Z_3)$, $H_{|1\rangle\langle 1|} = -\omega Z_1$; this yields a 2-local Hamiltonian $H'$ simulating the 3-local interaction $H = Z_1 \otimes Z_2 \otimes Z_3$. More generally, given three commuting Pauli strings $A_a$, $B_b$, $C_c$, we can set $H_{\mathrm{I}} = -A_a$, $H_X = \sqrt{\frac{\omega}{2}}(B_b + C_c)$, $H_{|1\rangle\langle 1|} = -\omega A_a$ to simulate the interaction $H = A_a \otimes B_b \otimes C_c$. This procedure may be used to simulate a $k$-local Pauli string using a $(\lceil k/3 \rceil + 1)$-local Hamiltonian.

Although Proposition 27 shows that evolution and repeated measurements under $H'$ reproduce the dynamics of $H$, it is also important to guarantee that it can be combined with other interactions. Proposition 28 provides the necessary result for this, by verifying that the conclusions of Proposition 27 also hold when an additional term $H_{\mathrm{else}} \in \mathrm{Herm}(\mathcal{H})$ is added to both the target and simulator Hamiltonian.

**Proposition 28.** Let $H_{\mathrm{else}} = \sum_i h_i$ be a $k$-local Hamiltonian on $\mathcal{H} = \otimes_i \mathcal{H}_i$ such that $\|h_i\| = O(1)$, and whose interaction graph has a degree bounded by an $O(1)$ constant.

Introduce an ancillary qubit $\mathcal{A} = \mathbb{C}^2$, and let $H' \in \mathrm{Herm}(\mathcal{H} \otimes \mathcal{A})$ be a Hamiltonian given by

$$H' = H_{\mathrm{I}} \otimes \mathrm{I} + H_X \otimes X + H_{|1\rangle\langle 1|} \otimes |1\rangle\langle 1|, \tag{57}$$

for some $H_{\mathrm{I}}, H_X, H_{|1\rangle\langle 1|} \in \mathrm{Herm}(\mathcal{H})$ depending on a small parameter $\delta t$ such that $\|H_{\mathrm{I}}\| = O(1)$, $\|H_X\| = O((\delta t)^{-1/2})$, and $\| H_{|1\rangle\langle 1|} \| = O((\delta t)^{-1})$ with $H_{|1\rangle\langle 1|}^2 = \omega^2 \mathrm{I}$, $\omega = \frac{2\pi}{\delta t}$. Assume that $H_{\mathrm{I}}, H_X$, and $H_{|1\rangle\langle 1|}$ act on $O(1)$ sites in $\mathcal{H}$.

Then, for any $|\psi\rangle \in \mathcal{H}$,

$$e^{-i\delta t(H' + H_{\mathrm{else}} \otimes \mathrm{I})}(|\psi\rangle \otimes |0\rangle) = \left(e^{-i\delta t(H + H_{\mathrm{else}})}|\psi\rangle + O((\delta t)^2)\right) \otimes |0\rangle + O((\delta t)^{3/2}) \otimes |1\rangle, \tag{58}$$

where

$$H = H_{\mathrm{I}} - \omega^{-2} H_X H_{|1\rangle\langle 1|} H_X. \tag{59}$$

The significance of Proposition 28 is that the errors do not depend on the size of the system through $\|H_{\mathrm{else}}\|$, due to bounds we place on the Trotter error in the expansion $e^{-i\delta t(H + H_{\mathrm{else}})} \approx e^{-i\delta t H} e^{-i\delta t H_{\mathrm{else}}}$.

## Discussion

Given the result of Proposition 28, we can now describe how the measurement gadget construction fits into our framework of analogue quantum simulation described in Definition 5.

Given a Hamiltonian $H = Z_1 \otimes Z_2 \otimes Z_3 + H_{\mathrm{else}}$ on $n$ qubits $\mathcal{H} = (\mathbb{C}^2)^{\otimes n}$, with $H_{\mathrm{else}} \in \mathrm{Herm}(\mathcal{H})$ satisfying the requirements of Proposition 28, we fix some $\delta t > 0$ and define the simulator space $\mathcal{H}' = \mathcal{H} \otimes \mathcal{A}$, where $\mathcal{A} = \mathbb{C}^2$. Let $H' \in \mathrm{Herm}(\mathcal{H}')$ be given by

$$H' = -Z_1 \otimes \mathrm{I} + \sqrt{\frac{\omega}{2}}(Z_2 + Z_3) \otimes X - \omega Z_1 \otimes |1\rangle\langle 1|, \tag{60}$$

where $\omega = \frac{2\pi}{\delta t}$. Define the state and observable encodings $\mathcal{E}_{\mathrm{state}}$ and $\mathcal{E}_{\mathrm{obs}}$ by

$$\mathcal{E}_{\mathrm{state}}(\rho) = \rho \otimes |0\rangle\langle 0|, \quad \mathcal{E}_{\mathrm{obs}}(O) = O \otimes \mathrm{I}, \tag{61}$$

and define channels $E_{\delta t}, M : D(\mathcal{H}') \to D(\mathcal{H}')$ by

$$E_{\delta t}(\rho') = e^{-i\delta t(H' + H_{\mathrm{else}} \otimes \mathrm{I})} \rho' e^{i\delta t(H' + H_{\mathrm{else}} \otimes \mathrm{I})}, \tag{62}$$

$$M(\rho') = \mathrm{tr}_{\mathcal{A}}[\rho'(\mathrm{I} \otimes |0\rangle\langle 0|)] \otimes |0\rangle\langle 0| + \mathrm{tr}_{\mathcal{A}}[\rho'(\mathrm{I} \otimes |1\rangle\langle 1|)] \otimes |1\rangle\langle 1|, \tag{63}$$

so that $E_{\delta t}$ corresponds to evolution under the Hamiltonian $H' + H_{\mathrm{else}}$ for time $\delta t$, and $M$ corresponds to a measurement of the $\mathcal{A}$ system. Then, for all $t$, define the time evolution channel

$$T_t = (M \circ E_{\delta t}) \circ (M \circ E_{\delta t}) \circ \cdots \circ (M \circ E_{\delta t}), \tag{64}$$

containing $\lfloor t/\delta t \rfloor$ copies of $(M \circ E_{\delta t})$. This evolution is described by Fig. 4. The content of Proposition 28 tells us that

$$(T_t \circ \mathcal{E}_{\mathrm{state}})(\rho) = \left(e^{-itH} \rho e^{itH} + O(t\delta t)\right) \otimes |0\rangle\langle 0| + O(t(\delta t)^2) \otimes |1\rangle\langle 1|, \tag{65}$$

and hence for any observable $O \in \mathrm{Herm}(\mathcal{H})$ with $\|O\|=1$,

$$\mathrm{tr}[\mathcal{E}_{\mathrm{obs}}(O)(T_t \circ \mathcal{E}_{\mathrm{state}})(\rho)] = \mathrm{tr}[Oe^{-itH}\rho e^{itH}] + O(t\delta t). \qquad (66)$$

The channels $T_t$ therefore simulate $H$ (in the sense of Definition 5) with respect to any states $\Omega_{\mathrm{state}}$ and normalised observables $\Omega_{\mathrm{obs}}$, up to accuracy $\epsilon > 0$ and maximum time $t_{\mathrm{max}}$, provided that one chooses $\delta t = O(\epsilon t_{\mathrm{max}}^{-1})$. Therefore we require interaction strengths and measurement frequency which scale as $J = O(\epsilon^{-1} t_{\mathrm{max}})$—note that this does not depend on $n$, the size of the system.

We can compare these scalings with those obtained if we were to use conventional gadgets. Suppose we have a $(\eta, \epsilon)$-gadget in the sense of Definition 15 with $\eta = O(n^{-1}\epsilon)$ to ensure an absolute error of $O(\epsilon)$ when combined with a Hamiltonian of order $n$, comparable with the above construction. By Theorem 26, this must involve energy scalings of $J = \Omega(\epsilon^{-1} n)$ (and even without Theorem 26, a low-energy $(\Delta, \eta, \epsilon)$-gadget as in Definition 18 would require energies scaling as $\Omega(n)$ to ensure that unwanted states are sufficiently penalised). In fact, this is likely not the optimal bound; the best known 3-to-2 gadget construction requires energy scales of $O(\epsilon^{-3} + \eta^{-3})$, which in this case would require interaction strengths scaling as $J = O((\epsilon^{-1}n)^3)$. Even if the system size is restricted via Lieb-Robinson bounds to set $n = O(\log^d(1/\epsilon) + t_{\mathrm{max}}^d)$ (where $d$ is the dimension of the system), the measurement-based gadget still provides an improvement.

Despite this advantage, the measurement gadget construction involves repeated instantaneous decoherence of the ancillary qubit at precise time intervals without disturbing the rest of the system, and may still require large (albeit non-scaling) interaction strengths. Moreover, if $N_{\mathrm{gad}}$ such gadgets were used in parallel, we expect (though do not calculate here) that an additional overhead of at least $\delta t = O((t_{\mathrm{max}} N_{\mathrm{gad}})^{-1/2})$ would be necessary to control the probability of measuring a 1 at any of the ancillary sites. Nonetheless, the construction provides a marked improvement in scalings over existing gadgets for a single 3-local term in a Hamiltonian, and gives some positive clues as to the ways in which simulators might take advantage of more general possibilities for channels allowed by Definition 5. We leave the detailed study of such gadgets, and their robustness to error for future work. We anticipate that, for a suitable adaptation of Definition 15 for the dissipative case, there may be similar no-go results preventing locality reduction by gadgets independently of the size of the system.

## Data availability
No datasets were generated or analysed during the current study.

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

## Acknowledgements

We acknowledge financial support from the Novo Nordisk Foundation (Grant No. NNF20OC0059939 'Quantum for Life'), the European Research Council (ERC Grant Agreement No. 818761) and VILLUM FONDEN via the QMATH Centre of Excellence (Grant No. 10059). A.H.W. thanks the VILLUM FONDEN for its support with a Villum Young Investigator Grant (Grant No. 25452). I.D. was supported in part by the AFOSR under grant FA9550-21-1-0392 and a National Science Foundation (NSF) Graduate Research Fellowship under Grant No. DGE 1656518. I.D. thanks everyone at QMATH for their hospitality during his research visit to KU and especially Prof. Adam Bouland for encouraging and supporting the visit. I.D. gratefully acknowledges Harriet Apel for generously offering insights and guidance during fruitful discussions at the early stages of this work.

## Author contributions

All authors discussed and made substantial contributions to the main ideas of the work. D.H. contributed the mathematical proofs and, with input from I.D., F.R.K., A.B., D.S.F., A.H.W., and M.C., drafted the manuscript. M.C. supervised the project.

## Competing interests

The authors declare no competing interests.

## Additional information

**Peer review information** : *Nature Communications* thanks the anonymous reviewer(s) for their contribution to the peer review of this work. A peer review file is available.

