## [Peer Review File · Nature Communications]

Going beyond gadgets: The importance of scalability for analogue quantum simulatorsREVIEWER COMMENTS

Reviewer #1 (Remarks to the Author):

In this work the authors extend the theoretical study of analogue Hamiltonian simulation, and attempt to bring it closer to experimental work. They argue that existing theoretical definitions of Hamiltonian simulation allow for physically unrealistic scalings of the interaction strengths in the simulator system, and moreover are unnecessarily demanding by requiring that the entire physics of a target system is captured in a simulator system, as opposed to just some particular class of observables.

The authors present a new definition of Hamiltonian simulation, which they argue is an improvement on previous definitions for two reasons:

- it explicitly requires that the simulation is size independent, so the interaction strengths in the simulator system do not depend on the size of the target system, and the size of the simulator system depends only linearly on the size of the target system
- it allows for simulations which only reproduce subsets of observables, and allows the map from target to simulator system to be a general quantum channel

The authors provide substantial evidence of the utility of their new definition: they demonstrate that a particular (wide) class of simulations under the previous definition will always have size dependent interaction strengths in the simulator, which they argue is infeasible. Moreover, they demonstrate that using their new definition of analogue simulation they are able to bypass their own no go result, by making use of dissipative processes, to construct a simulation which does not depend on system size. The simulation technique they derive does have some of its own drawbacks, but these are well covered in the text (e.g. need for rapid measurement of parts of a system without disturbing the rest of the system).

Overall I think this is a very interesting and well put together paper, which could play a role in the important task of bringing theoretical and experimental work on analogue simulation

closer together. I would recommend it for acceptance in Nature Communications. I have two minor comments for the authors:

1. One of the key benefits of the previous definition of analogue Hamiltonian simulation (from ref. 12) is that it supports the arguments that error correction is not needed for analogue simulation because the errors in the simulator system map to 'natural noise processes' in the system being simulated, and therefore the errors 'form part of the simulation'. Do you know if that is also true for your new definition of simulation? There is a brief discussion of noise on page 7 of the current manuscript, but it seems to just be saying that if the noise is small then the simulation is approximately correct - the previous results were stronger here. It may be that this is not a straightforward question to answer, in that case I wouldn't say the paper had to include an answer before it could be published, but I would be interested to know if the authors have thought about it or not. It may be a quick addition to the paper, or it might be an interesting avenue for a follow up paper.

2. I like the layout of the paper with all the proofs relegated to the end, it makes the argument clear and easy to read. However, it would be useful to add hyperlinks after theorems / lemmas / etc. to the place in the paper where they are proven, to make it easier to follow the details if readers want to.

Reviewer #2 (Remarks to the Author):

The authors examine the formalism of analogue Hamiltonian simulation and find a limitation in the unphysical system size scaling of interaction strengths required by some constructions, most notably the perturbative gadgets that are used to reduce the support of Hamiltonian terms. They find similar issues for the low energy simulation of qudit Hamiltonians by means of qubit Hamiltonians.

In view of this, they propose an alternative, generalized framework for analogue quantum simulation which, even if it does not by itself lead to the exclusion of unphysical scalings of interaction strengths, it permits to circumvent them by (i) allowing for restricted application of the simulator to specific subspaces of initial states and operators, and (ii) allowing for

more general time evolution in the simulator. Ultimately the latter generalization is used to define simulators that operate via dissipative evolution, freezing the ancillary degrees of freedom to their target subspace by means of repeated measurement, as opposed to strong interactions. These dissipative gadgets are proposed as an alternative, with its own pros and cons, to the original ones, based on unitary evolution. In that sense, I'd say a more appropriate title would be "Going beyond unitary gadgets", but I do not find it an imprecision worth a title change.

The authors have identified and proposed a way of solving a problem in analogue quantum simulation. It is true that, from the perspective of motivating the work, it is not clear whether the constructions displaying unphysical scaling of interaction strengths are purely theoretical or whether they are expected to be implemented in actual experiments: a few comments along those lines would be welcome. In any case, the authors deal with the problem in a decently rigorous and systematic manner, building a general mathematical framework for gadgets on top of previous proposals, within which they prove their results. The idea of using dissipation to aid in quantum information processing is not new, and the term dissipative gadget has been used in other more or less related contexts. Nevertheless, the construction presented in this paper and its motivation as a way to remedy the scaling of interaction strengths, as well as the structured formalism that accompanies the proposal and the no-go theorems are relevant contributions from my point of view. I would add that the no-go results and the proposed dissipative gadgets are rather specific, proof-of-principle examples and it would be nice to have a few additional sentences describing the outlook of how the authors think their proposals can be extended, generalized and applied to other problems.

The following are some other comments of varying degree of importance regarding the results and presentation (some of them may just reflect my shortcomings in understanding the proofs):

1. To avoid confusion, I would specify that the norm to be represented by $|| \cdot ||$ during the whole paper is the operator norm.
2. The paragraph right after Definition 1 is repeated verbatim from the introduction of the paper.

3. The unfamiliar reader meets Definition 3 and 4 with very little motivation. Even if it is going to be superseded (or precisely because it will), it would be nice to know the reasons this approach was proposed. Also the bar notation for the complex conjugate in Def. 3 should be explained.
4. I find the explanation for considering the encoding of operators to be a unital CP map (page 5, second paragraph) a bit confusing, since I do not understand the meaning of “accommodating quantum outputs”.
5. Proposition 6 is addressed as Theorem earlier in the text.
6. Is there a reason α is upper bounded by 1 in Proposition 6?
7. The notation $*$ for the dual channel in the “Noisy analogue simulators” section should be explained.
8. Right before section 4, the t^d should be t_{\max}^d .
9. The presentation (and proof) of Prop. 19 was a bit confusing for me. It seems that ϵ , η , Δ are properties of the individual gadgets, so why do they scale with n when combined?
10. Right before Proposition 22, it is emphasized that dt should scale as $1/\sqrt{t}$ to control the probability of measurement success, but it seems this is superseded by the scaling needed to control the error in case of measurement success, which should be $1/t$.
11. The last equation of page 18 is missing the observable O .
12. Before the equation before (16), “so there exists a projector $P\dots$ ”: this statement is not obvious to me, nor is the later fact that it has the same rank as \tilde{P} .
13. After (20), it is not intuitive to me why we can do the redefinition by a multiple of the identity, it would be nice to be more explicit there.
14. I find the inductive argument of the proof of Theorem 12 may benefit from a more detailed presentation. For instance, right after (21), $H^{(k-1)}$ is restricted to a subspace which, if I am not mistaken, it does not leave invariant. Also, the comment at the end of the page seems to require “sufficiently small η and ϵ ”, of which no mention is made in the statement of the theorem. Later in the proof, the statement that “the null space of $P'H'P'$ is exactly that of P' ” is also not obvious to me.
15. As a suggestion, now that I am typing this, numbering all equations even if one does not cite them in the text makes referencing parts of the paper easier.
16. Right before the first bullet point of page 30, I'd say the statement that $I \otimes P_j$

commutes with H and S could use more explaining. At the end of that page, it is unclear to me which is the $O(\eta^2 \eta)$ term.

17. The proof of Theorem 19 also felt a bit confusing for me. The statements about block-diagonality in page 32 are a bit unclear to me, and it feels like J sometimes means J' . Also the equation after (25) is missing the closing norm $||$.

18. I am confused by the statement after (28), where n can be chosen to make ω small enough, since the right hand side of the inequality also scales with n , does it not?

19. I do not know where the lower bound on Δ comes from after (29).

20. On page 37, Lemma 35 is invoked but F does not seem to be the same k -local function for which the lemma was proved, unless $k' = 0$.

21. Right after (41), the derivative of f_X is missing an i (or an absolute value).

Based on the above, once the comments I have made have been sufficiently addressed, I would recommend publication in Nature Communications.

Reviewer #3 (Remarks to the Author):

This manuscript studies the resource requirement for analogue quantum simulation, which is an important application for quantum technology. Previously, theoretical foundation of analogue quantum simulators has been laid in Hamiltonian complexity theory, where various papers have described constructive mappings between quantum systems and established existence of universal families of Hamiltonians that can simulate all other systems. Nevertheless, one key issue remains: Existing approaches for simulating arbitrary quantum systems generally require interaction strengths in the simulator that diverge with the target system size. Since physical systems usually have bounded interaction energy, removing such a requirement would allow for more scalable implementation of analogue simulators.

Motivated by this perspective, this paper advocates for the study of "size-independent simulation", where the simulator has bounded interaction strength and at most linear overhead in the number of qubits. In this context, the paper contains the following main results:

1) The paper proposes an alternative definition of analogue quantum simulators that only requires simulation of dynamics of observable. This is in contrast to previous, commonly used definitions (e.g. by Cubitt et al.) that requires the simulator to reproduce the full spectrum of the target system. The paper argues the previous definition has certain shortcomings that fundamentally limit scalability; for example, an overhead that diverges with system size is necessary even for a simple task of simulating a system of non-interacting qutrits using qubits. This is not a problem with the new definition that has more relaxed requirements.

2) The paper also proposes a general formulation of Hamiltonian gadgets, which have been previously used in analogue simulations. This formalism is used to prove that the interaction strengths would necessarily diverge with system sizes when such gadgets are used to reduce locality of interactions for the simulator.

3) Lastly, the paper proposes a new type of gadget that uses engineered dissipation to achieve a more favorable resource scaling under the new definition of analogue quantum simulation of dynamics.

In my view, I think these constitute a nice set of results that are very interesting for the field of analogue quantum simulation and Hamiltonian complexity theory. I like the paper's insight that the current notions of analogue simulation may have fundamental limitation that prevents scalability. The rigorous lower bounds in the paper on the required interaction strengths for various Hamiltonian reduction tasks are also valuable theoretical contributions. Additionally, I think the proposed idea of using open-system dynamics for analogue simulation points to a promising direction that could be impactful for both theory and practice. Nevertheless, I have a few reservations about these results that somewhat undercut the paper's message:

1) While the bounded interaction requirement is physically motivated, I think the paper's requirement in "size-independent simulation" that the number of qubits in the simulator scales at most linearly is a bit unnecessarily strong. I think it's far easier for the experimentalist to add more qubits than to amplify the interaction. Without the

requirement of linear overhead, the motivating example (in Proposition 6) for the new definition of seems to fail since one can set $\alpha = 1$ and get merely a constant lower bound on the interaction strength. I wonder, is there a way to motivate the new definition without requiring linear overhead in qubit number?

2) The proposed new definition of analogue quantum simulator might be too broad to be useful. For example, a digital quantum simulator would also qualify under the new definition. Less trivially, Hamiltonian simulators of quantum circuits (e.g. the quantum walk Hamiltonians by Childs, Gosset, and Webb, or Hamiltonian Quantum Cellular Automata by Nagaj and Wocjan) would also qualify as analogue simulators. Moreover, they satisfy the requirement of bounded interaction strength which I believe is the main goal of this paper.

3) While the proposed new gadget using quantum Zeno effect very nicely improves the resource scaling, I worry that the level of control required may make it more susceptible to errors, negating a key benefit of analogue quantum simulators. Specifically, it seems to require the implementation of fine-tuned interaction strength in the Hamiltonian (ω) as well as evolution duration (Δt). At this level, the proposal almost seems to have similar requirements as digital quantum simulators. In contrast, in previously studied notions of analogue simulator, small imperfections arguably doesn't significantly affect important properties of the simulation, and local noise in the simulator maps to local noise in the target system. It would be nice if the paper also includes a closer look at how errors propagate in these new types of gadgets, and potentially some argument of robustness.

Since the main message of the paper seems to be that we need a new definition of simulation and new notions of simulation gadgets, I think the paper could benefit from more carefully arguing for why these new notions are necessary, and better clarification of the caveats and limitations. Nevertheless, I find the manuscript generally well-written, the theoretical analysis sound, and the results conceptually and technically interesting and potentially impactful. Assuming the authors are able to address my concerns in a revised manuscript, I would be happy to recommend its publication.

Some additional comments:

1) Third paragraph of main text, where the paper claims to have proven that encoding n non-interaction qutrits into qubits require interaction scaling as n . I think it would be better if the author clarifies the caveat of this result, namely the additional requirement that the qubit overhead scales at most linearly.

2) Fourth paragraph: The civil engineer analogy does not fully resonate with me, since I think it's not unnatural for a simulator to require higher precision in the building blocks when the system size is larger.

3) In Figure 3(b), the theorem & proposition numbering are not all correct. Also, it's a bit unclear what this figure is trying to say, especially since the items in the boxes are not obviously connected to discussions in the main text.

4) The explanation of the gadget formalism in the main text is a bit confusing. The reader could benefit some text clarifying connection between η and ζ and the intuition behind them. In particular, it is not clear why η must decrease with the size of the system.

Dear Referees,

We thank you for the thoughtful and useful comments. We have made edits to the manuscript where appropriate, and attach a revised version, as well as a version with edits highlighted and a list of changes. On the following pages the reviewer comments are copied out in full, and we have added responses and annotations in blue.

Reviewer 1

In this work the authors extend the theoretical study of analogue Hamiltonian simulation, and attempt to bring it closer to experimental work. They argue that existing theoretical definitions of Hamiltonian simulation allow for physically unrealistic scalings of the interaction strengths in the simulator system, and moreover are unnecessarily demanding by requiring that the entire physics of a target system is captured in a simulator system, as opposed to just some particular class of observables.

The authors present a new definition of Hamiltonian simulation, which they argue is an improvement on previous definitions for two reasons:

- it explicitly requires that the simulation is size independent, so the interaction strengths in the simulator system do not depend on the size of the target system, and the size of the simulator system depends only linearly on the size of the target system
- it allows for simulations which only reproduce subsets of observables, and allows the map from target to simulator system to be a general quantum channel

The authors provide substantial evidence of the utility of their new definition: they demonstrate that a particular (wide) class of simulations under the previous definition will always have size dependent interaction strengths in the simulator, which they argue is infeasible. Moreover, they demonstrate that using their new definition of analogue simulation they are able to bypass their own no go result, by making use of dissipative processes, to construct a simulation which does not depend on system size. The simulation technique they derive does have some of its own drawbacks, but these are well covered in the text (e.g. need for rapid measurement of parts of a system without disturbing the rest of the system).

Overall I think this is a very interesting and well put together paper, which could play a role in the important task of bringing theoretical and experimental work on analogue simulation closer together. I would recommend it for acceptance in Nature Communications. I have two minor comments for the authors:

1. One of the key benefits of the previous definition of analogue Hamiltonian simulation (from ref. 12) is that it supports the arguments that error correction is not needed for analogue simulation because the errors in the simulator system map to ‘natural noise processes’ in the system being simulated, and therefore the errors ‘form part of the simulation’. Do you know if that is also true for your new definition of simulation? There is a brief discussion of noise on page 7 of the current manuscript, but it seems to just be saying that if the noise is small then the simulation is approximately correct - the previous results were stronger here. It may be that this is not a straightforward question to answer, in that case I wouldn’t say the paper had to include an answer before it could be published, but I would be interested to know if the authors have thought about it or not. It may be a quick addition to the paper, or it might be an interesting avenue for a follow up paper.

We have added a short section (3.1) in the supplementary information, which contains some definitions for sensible “local” encodings of both states and operators, as well as some basic results. The main points are as follows:

- We define local state encodings to be those implementable by a shallow-depth quantum circuit (a condition that can equivalently be stated in terms of having a Stinespring unitary which maps local operators to local operators).
- This gives an analogous result to the case of [CMP18] in terms of the back-propagation of local errors.

- We also define local observable encodings, which are defined to be Hilbert-Schmidt adjoints of local state encodings. Such encodings have the property that measurement of an encoded local observable can be done via a local POVM on the simulator.

The section also includes a reference to [TRC22], where some related results for noisy analogue quantum simulation of observables are proved.

2. I like the layout of the paper with all the proofs relegated to the end, it makes the argument clear and easy to read. However, it would be useful to add hyperlinks after theorems / lemmas / etc. to the place in the paper where they are proven, to make it easier to follow the details if readers want to.

Hyperlinks have now been added to theorems, proofs, references, hopefully this improves readability.

Reviewer 2

The authors examine the formalism of analogue Hamiltonian simulation and find a limitation in the unphysical system size scaling of interaction strengths required by some constructions, most notably the perturbative gadgets that are used to reduce the support of Hamiltonian terms. They find similar issues for the low energy simulation of qudit Hamiltonians by means of qubit Hamiltonians.

In view of this, they propose an alternative, generalized framework for analogue quantum simulation which, even if it does not by itself lead to the exclusion of unphysical scalings of interaction strengths, it permits to circumvent them by (i) allowing for restricted application of the simulator to specific subspaces of initial states and operators, and (ii) allowing for more general time evolution in the simulator. Ultimately the latter generalization is used to define simulators that operate via dissipative evolution, freezing the ancillary degrees of freedom to their target subspace by means of repeated measurement, as opposed to strong interactions. These dissipative gadgets are proposed as an alternative, with its own pros and cons, to the original ones, based on unitary evolution. In that sense, I'd say a more appropriate title would be "Going beyond unitary gadgets", but I do not find it an imprecision worth a title change.

The authors have identified and proposed a way of solving a problem in analogue quantum simulation. It is true that, from the perspective of motivating the work, it is not clear whether the constructions displaying unphysical scaling of interaction strengths are purely theoretical or whether they are expected to be implemented in actual experiments: a few comments along those lines would be welcome.

Of course, gadgets so far have been primarily used as a theoretical tool — though there have been attempts to design gadgets for more practical situations [CBBK13, CFKE22]. However as our work shows, it may be impossible to build practical gadgets in many situations. The second paragraph of the "generalising Hamiltonian gadgets" section of the main text has been edited to clarify this.

In any case, the authors deal with the problem in a decently rigorous and systematic manner, building a general mathematical framework for gadgets on top of previous proposals, within which they prove their results. The idea of using dissipation to aid in quantum information processing is not new, and the term dissipative gadget has been used in other more or less related contexts.

The final paragraph of main text has been edited to note existing theoretical and practical uses of the quantum Zeno effect, in particular citing [BMD⁺22, LMF⁺23].

Nevertheless, the construction presented in this paper and its motivation as a way to remedy the scaling of interaction strengths, as well as the structured formalism that accompanies the proposal and the no-go theorems are relevant contributions from my point of view. I would add that the no-go results and the proposed dissipative gadgets are rather specific, proof-of-principle examples and it would be nice to have a few additional sentences describing the outlook of how the authors think their proposals can be extended, generalized and applied to other problems.

Indeed, the example given is primarily intended as a proof-of-principle showing that the gadget energy scaling theorem can be somewhat circumvented. It would be interesting to analyse this more deeply, and hopefully generalise the construction, but we leave this for future work — a couple of sentences to this effect have been added to the final paragraph of section 5.

The following are some other comments of varying degree of importance regarding the results and presentation (some of them may just reflect my shortcomings in understanding the proofs):

1. To avoid confusion, I would specify that the norm to be represented by $\|\cdot\|$ during the whole paper is the operator norm.

Clarification has been added to main text, at the "combination" bullet point.

2. The paragraph right after Definition 1 is repeated verbatim from the introduction of the paper.

This paragraph has been removed, we thank the referee for pointing out the error.

3. The unfamiliar reader meets Definition 3 and 4 with very little motivation. Even if it is going to be superseded (or precisely because it will), it would be nice to know the reasons this approach was proposed. Also the bar notation for the complex conjugate in Def. 3 should be explained.

A few more explanatory words have been added before Definition 3: “By deriving the most general possible form of a spectrum-preserving Hamiltonian encoding, and then imposing natural locality conditions,” . The use of the complex conjugate notation has also now been clarified.

4. I find the explanation for considering the encoding of operators to be a unital CP map (page 5, second paragraph) a bit confusing, since I do not understand the meaning of “accommodating quantum outputs”.

Requiring \mathcal{E}_{obs} to be unital and CP is equivalent to requiring its Hilbert-Schmidt dual $\mathcal{E}_{\text{obs}}^*$ to be a CPTP quantum channel. Therefore, we can view the encoded observable expectation value

$$\text{tr}[\mathcal{E}_{\text{obs}}(O)\rho'] = \text{tr}[O\mathcal{E}_{\text{obs}}^*(\rho')]$$

as the expectation of the original observable O on a “decoded” state $\mathcal{E}_{\text{obs}}^*(\rho')$. An extra sentence has been added to the paragraph in question to clarify this.

5. Proposition 6(now 10 after relabelling) is addressed as Theorem earlier in the text. This has been fixed.

6. Is there a reason alpha is upper bounded by 1 in Proposition 6(now 10 after relabelling)?

The proposition was stated for $\alpha \in [0, 1]$ just because, if $\alpha \geq 1$, the bound given by the proposition is essentially trivial. We have now edited the statement to say $\alpha \geq 0$, though, to make it clearer that there is no real problem with making α larger.

7. The notation $*$ for the dual channel in the “Noisy analogue simulators” section should be explained.

We have added clarification that this represents the dual with respect to the Hilbert-Schmidt inner product.

8. Right before section 4, the t^d should be t_{max}^d .

This has been corrected.

9. The presentation (and proof) of Prop. 19(now 23 after relabelling) was a bit confusing for me. It seems that epsilon, eta, Delta are properties of the individual gadgets, so why do they scale with n when combined?

We have added clarification to Setup 21 (which used to be Setup 17). Formally, the propositions are talking about a family of systems, one for each n , and each with a set of gadgets with ϵ, η, Δ parameters (depending on n). If the gadgets have constant ϵ, η, δ with n , then eventually n will become large enough that the gadgets don’t combine properly. Thus ϵ, η, Δ must scale with n . But indeed, it is not individual gadgets that are scaling, the point is that better gadgets are needed for larger systems.

10. Right before Proposition 22(now 26 after relabelling), it is emphasized that δt should scale as $1/\sqrt{t}$ to control the probability of measurement success, but it seems this is superseded by the scaling needed to control the error in case of measurement success, which should be $1/t$.

Indeed, if δt is taken to be $O(t^{-1})$, then the condition will automatically be satisfied. A few more words have been added just before the proposition to address this: “which will always be...”

11. The last equation of page 18 is missing the observable O .

This has been corrected.

12. Before the equation before (16), “so there exists a projector $P\dots$ ”: this statement is not obvious to me, nor is the later fact that it has the same rank as \tilde{P} .

We have added some text to clarify this, and fixed a factor of two in the error bound.

The point is that we have shown $\|P' - \mathbb{I} \otimes M\| \leq \delta$ (where $M = K^{-1} \text{tr}_{\mathcal{H}}[P']$ and $\delta = \pi K \sqrt{\zeta}$). Since M is δ -far from a projector, it must have eigenvalues which are at most δ away from 0 and 1 by Weyl’s perturbation theorem. Therefore we construct P by just rounding all of the eigenvalues of M to the nearest integer. We then have

$$\|P' - \mathbb{I} \otimes P\| \leq \|P' - \mathbb{I} \otimes M\| + \|P - M\| \leq 2\delta .$$

The claim about the rank follows from the fact that we know

$$\|P'(H_{\text{else}} \otimes \mathbb{I})P' - \tilde{U}_{\lambda H_{\text{else}}}(H_{\text{else}} \otimes \tilde{P})\tilde{U}_{\lambda H_{\text{else}}}^\dagger\| \leq \zeta + O(\lambda^{-1}) ,$$

for all H_{else} . Therefore, for small enough ζ , we have $\text{rank}(\mathbb{I} \otimes P) = \text{rank} P' = \text{rank}(\mathbb{I} \otimes \tilde{P})$. Text has been added to clarify this too.

13. After (20), it is not intuitive to me why we can do the redefinition by a multiple of the identity, it would be nice to be more explicit there.

A few more lines have been added to the proof to clarify this. The point is that we can replace $H \mapsto H + \mu \mathbb{I}$, $H' \mapsto H' + \mu \mathbb{I}$, and these also satisfy the assumptions of the theorem with an adjusted ϵ . If we can prove that $(H' + \mu \mathbb{I})$ and $(H + \mu \mathbb{I})$ satisfy the gadget definition, then so do the original H' and H .

14. I find the inductive argument of the proof of Theorem 12 may benefit from a more detailed presentation. For instance, right after (21), $H^{(k-1)}$ is restricted to a subspace which, if I am not mistaken, it does not leave invariant.

$H^{(k-1)}$ is constructed so that its first $(k-1)$ eigenspaces have been rotated onto $\mathcal{P}'_1, \mathcal{P}'_2, \dots, \mathcal{P}'_{k-1}$. So it leaves the space $\oplus_{j < k} \mathcal{P}'_j$ invariant. Since it’s Hermitian, it also leaves $\oplus_{j \geq k} \mathcal{P}'_j$ invariant, so the restriction is well-defined. Some text has been added to clarify this.

Also, the comment at the end of the page seems to require “sufficiently small eta and epsilon”, of which no mention is made in the statement of the theorem.

This was a typo, the η should have been a ζ — it has now been fixed. Since the theorem is stated in terms of quantities shrinking asymptotically with ζ and ϵ , we can without loss of generality assume that they are small.

Later in the proof, the statement that “the null space of $P'H'P'$ is exactly that of P' “ is also not obvious to me.

By (17), we have $\text{rank} P' = \text{rank}(\mathbb{I} \otimes P)$. Since we shifted H by a factor of the identity to have only positive eigenvalues, we have $\text{rank}(\mathbb{I} \otimes P) = \text{rank}(H \otimes P)$. By (18), we have $\text{rank}(H \otimes P) = \text{rank}(P'H'P')$. Putting this all together gives

$$\text{rank} P' = \text{rank}(\mathbb{I} \otimes P) = \text{rank}(H \otimes P) = \text{rank}(P'H'P') ,$$

so the null space of P' , which is certainly a subspace of the null space of $P'H'H'$, is in fact the whole thing.

Explanation has been added of these steps in the manuscript.

15. As a suggestion, now that I am typing this, numbering all equations even if one does not cite them in the text makes referencing parts of the paper easier.

We have now added numbering to all display equations.

16. Right before the first bullet point of page 30, I'd say the statement that $I \otimes P_j$ commutes with H and S could use more explaining.

This is because P_j is acting only on the ancillary \mathcal{A}_j system, whilst H_k and S_k act on $\mathcal{H} \otimes \mathcal{A}_k$. Some text has been added to clarify this.

At the end of that page, it is unclear to me which is the $O(\eta^2\eta)$ term.

This was a typo, it should have read $O(\eta^3\epsilon)$. (The 3 factors of η come from the $\|S_j\|\|S_k\|\|S_l\|$, and the factor of ϵ drops out of what is now Lemma 35.) It has now been fixed.

17. The proof of Theorem 19 also felt a bit confusing for me. The statements about block-diagonality in page 32 are a bit unclear to me,

We thank the referee for raising this; it turned out there was an incorrect statement about block-diagonality made (in particular the “but note that, if $j \neq i$, then $[S_j, \tilde{H}_i]$ is ...”), which was a small hole in the proof. This has now been rectified. The new argument adds new terms to ω of order $O(\eta J + \eta\epsilon)$, but this doesn't make a difference to the conclusions of the theorem.

and it feels like J sometimes means J' .

This was perhaps due to two typos just above (28), where J was written instead of J' . One of these equations has now been removed due to the above point, and the other has been fixed.

Also the equation after (25) is missing the closing norm $\|$.

This has been corrected.

18. I am confused by the statement after (28), where n can be chosen to make ω small enough, since the right hand side of the inequality also scales with n , does it not?

This was poorly justified, and has now been fixed. An extra line has been added to the start of the proof to note $\tau/\Delta \rightarrow 0$ is implied by the assumptions. This, along with the other scaling assumptions in the theorem, is enough to deduce that for large n we have $\omega < \frac{1}{2}\Delta - (\|H\| - \tau)$. See edited manuscript for details.

19. I do not know where the lower bound on Δ comes from after (29).

This follows from the low-energy gadget definition (now Definition 17), from which we are assuming that $P_{\leq \Delta}(H'_i)H'_iP_{\leq \Delta}(H'_i)$ has the same spectrum as H_i , up to error ϵ . This means that the $(\leq \Delta)$ -energy subspace of H'_i must have an eigenvalue at least as big as $\|H_i\| - \epsilon$, so $\Delta \geq \|H_i\| - \epsilon$. Some text has been added to clarify this.

20. On page 37, Lemma 35 is invoked but F does not seem to be the same k -local function for which the lemma was proved, unless $k' = 0$.

The proof of the lemma has been rewritten in terms of the parity function for consistency. In fact, it turns out that this both simplifies the proof and strengthens the result, getting rid of the $2^{-k'}$ factor! So the statement and proof of the gadget energy scaling theorem (Theorem 25) have also been updated accordingly.

21. Right after (41), the derivative of f_X is missing an i (or an absolute value).

This has now been corrected.

Based on the above, once the comments I have made have been sufficiently addressed, I would recommend publication in Nature Communications.

Reviewer 3

This manuscript studies the resource requirement for analogue quantum simulation, which is an important application for quantum technology. Previously, theoretical foundation of analogue quantum simulators has been laid in Hamiltonian complexity theory, where various papers have described constructive mappings between quantum systems and established existence of universal families of Hamiltonians that can simulate all other systems. Nevertheless, one key issue remains: Existing approaches for simulating arbitrary quantum systems generally require interaction strengths in the simulator that diverge with the target system size. Since physical systems usually have bounded interaction energy, removing such a requirement would allow for more scalable implementation of analogue simulators.

Motivated by this perspective, this paper advocates for the study of “size-independent simulation”, where the simulator has bounded interaction strength and at most linear overhead in the number of qubits. In this context, the paper contains the following main results:

- 1) The paper proposes an alternative definition of analogue quantum simulators that only requires simulation of dynamics of observable. This is in contrast to previous, commonly used definitions (e.g. by Cubitt et al.) that requires the simulator to reproduce the full spectrum of the target system. The paper argues the previous definition has certain shortcomings that fundamentally limit scalability; for example, an overhead that diverges with system size is necessary even for a simple task of simulating a system of non-interacting qubits using qubits. This is not a problem with the new definition that has more relaxed requirements.
- 2) The paper also proposes a general formulation of Hamiltonian gadgets, which have been previously used in analogue simulations. This formalism is used to prove that the interaction strengths would necessarily diverge with system sizes when such gadgets are used to reduce locality of interactions for the simulator.
- 3) Lastly, the paper proposes a new type of gadget that uses engineered dissipation to achieve a more favorable resource scaling under the new definition of analogue quantum simulation of dynamics.

In my view, I think these constitute a nice set of results that are very interesting for the field of analogue quantum simulation and Hamiltonian complexity theory. I like the paper’s insight that the current notions of analogue simulation may have fundamental limitation that prevents scalability. The rigorous lower bounds in the paper on the required interaction strengths for various Hamiltonian reduction tasks are also valuable theoretical contributions. Additionally, I think the proposed idea of using open-system dynamics for analogue simulation points to a promising direction that could be impactful for both theory and practice. Nevertheless, I have a few reservations about these results that somewhat undercut the paper’s message:

- 1) While the bounded interaction requirement is physically motivated, I think the paper’s requirement in “size-independent simulation” that the number of qubits in the simulator scales at most linearly is a bit unnecessarily strong. I think it’s far easier for the experimentalist to add more qubits than to amplify the interaction. Without the requirement of linear overhead, the motivating example (in Proposition 6(now 10 after relabelling)) for the new definition of seems to fail since one can set $\alpha = 1$ and get merely a constant lower bound on the interaction strength. I wonder, is there a way to motivate the new definition without requiring linear overhead in qubit number?

Although the requirement of adding more qubits is not necessarily as much of a physical challenge for the experimentalist in itself, this may introduce problems elsewhere in the simulation. If each qubit is encoded into more than a constant number, then it will be impossible to fit them into a space of the same dimension without sacrificing geometric locality (a couple of sentences have been

added just before Definition 1 along these lines). The new section on local encodings also helps to motivate this: encoding with geometric locality is necessary in order to ensure that reasonable errors on the simulator correspond to reasonable errors on the target system, and that local observables can be measured locally on the target system.

Additionally, the $\alpha = 1$ case corresponds to a rather bizarre situation — using $\Omega(n^2)$ qubits to simulate a *non-interacting* system of size n is unlikely to be seen as reasonable by the experimentalist, even if it is possible! The paragraph after the proposition has been edited to reflect this.

- 2) The proposed new definition of analogue quantum simulator might be too broad to be useful. For example, a digital quantum simulator would also qualify under the new definition. Less trivially, Hamiltonian simulators of quantum circuits (e.g. the quantum walk Hamiltonians by Childs, Gosset, and Webb, or Hamiltonian Quantum Cellular Automata by Nagaj and Wocjan) would also qualify as analogue simulators. Moreover, they satisfy the requirement of bounded interaction strength which I believe is the main goal of this paper.

We have added a few sentences addressing this at the end of the “the analogue quantum simulator” section of the main text, and just before Section 3.1 in the supplementary text.

In particular, we have mentioned that the T_t should be implementable without the use of feed-forward measurements, in order to prevent error correction — a key distinguishing feature from digital quantum computation.

For quantum walk Hamiltonians, there are a few reasons they are not consistent with the notion of simulation we define. Firstly, encoding from a Hamiltonian into a digital simulation circuit, and then into a quantum walk, would incur a superlinear scaling in the size of the system. The necessary evolution time would also scale with the size of the system — if we interpret T_t as being a channel which really takes time t to implement, then this corresponds to a polynomial scaling of interaction strengths in order to normalise the time. More fundamentally, the definition we give requires a static observable encoding \mathcal{E}_{obs} which works for all $t \in [0, t_{\text{max}}]$. However, a quantum walk Hamiltonian would have the computation “moving through” the system, so it is not clear how to extract observable expectations at intermediate times with a constant and local \mathcal{E}_{obs} .

- 3) While the proposed new gadget using quantum Zeno effect very nicely improves the resource scaling, I worry that the level of control required may make it more susceptible to errors, negating a key benefit of analogue quantum simulators. Specifically, it seems to require the implementation of fine-tuned interaction strength in the Hamiltonian (ω) as well as evolution duration (δt). At this level, the proposal almost seems to have similar requirements as digital quantum simulators. In contrast, in previously studied notions of analogue simulator, small imperfections arguably doesn’t significantly affect important properties of the simulation, and local noise in the simulator maps to local noise in the target system. It would be nice if the paper also includes a closer look at how errors propagate in these new types of gadgets, and potentially some argument of robustness.

The main purpose of the dissipative gadget construction was as a proof-of-principle to highlight that the gadget energy scaling result could be somewhat circumvented in the non-unitary setting — but we acknowledge that it introduces new experimental issues (second paragraph at the start of Section 5 has been edited to emphasise this).

Certainly, it would be interesting to try to generalise the dissipative constructions and analyse the propagation of errors, but this is a nontrivial task and we think it is outside of the scope of this work. Indeed, we expect that there are likely other theoretical barriers that will prevent even dissipative gadgets from achieving fully scalable and error-robust implementation. Words to this effect have been added at the end of section 5.

Since the main message of the paper seems to be that we need a new definition of simulation and new notions of simulation gadgets, I think the paper could benefit from more carefully arguing for why these new notions are necessary, and better clarification of the caveats and limitations. Nevertheless, I find the manuscript generally well-written, the theoretical analysis sound, and the results conceptually and technically interesting and potentially impactful. Assuming the authors are able to address my concerns in a revised manuscript, I would be happy to recommend its publication.

Some additional comments:

- 1) Third paragraph of main text, where the paper claims to have proven that encoding n non-interaction qutrits into qubits require interaction scaling as n . I think it would be better if the author clarifies the caveat of this result, namely the additional requirement that the qubit overhead scales at most linearly.

Clarification has been added to this paragraph.

- 2) Fourth paragraph: The civil engineer analogy does not fully resonate with me, since I think it's not unnatural for a simulator to require higher precision in the building blocks when the system size is larger.

The text has been edited slightly to include “rather than just the density of the bricks in the original system”. Hopefully this emphasises the point that the scalings aren't reflecting anything physical in the original system; rather, they are building blocks which must all be adjusted every time a new building block is added, violating the concept of building blocks!

- 3) In Figure 3(b), the theorem & proposition numbering are not all correct. Also, it's a bit unclear what this figure is trying to say, especially since the items in the boxes are not obviously connected to discussions in the main text.

We thank the referee for pointing out this error — the numbering in Fig. 3b has now been corrected. Also some additional detail has been added to the caption to clarify the relationship of the results to the main text.

- 4) The explanation of the gadget formalism in the main text is a bit confusing. The reader could benefit some text clarifying connection between η and ζ and the intuition behind them. In particular, it is not clear why η must decrease with the size of the system.

We have added text in parentheses “(where η ...)”, and a sentence “In particular, when $\|H_{\text{else}}\| \sim n$...” to clarify this. Essentially, η and ζ are morally measuring the same thing: the ability of the gadget to combine with other terms — if one of them is small, then the other one is too (unfortunately not linearly). In order to offset an increase in the magnitude of the Hamiltonian $\|H_{\text{else}}\|$, η must therefore decrease to control the error.

References

- [BMD⁺22] Eliya Blumenthal, Chen Mor, Asaf A. Diring, Leigh S. Martin, Philippe Lewalle, Daniel Burgarth, K. Birgitta Whaley, and Shay Hacohe-Gourgy. Demonstration of universal control between non-interacting qubits using the Quantum Zeno effect. *npj Quantum Information*, 8(1):88, 2022.
- [CBBK13] Yudong Cao, Ryan Babbush, Jacob Biamonte, and Sabre Kais. Towards experimentally realizable Hamiltonian gadgets. *ArXiv e-prints*, 1311, 2013.
- [CFKE22] Simon Cichy, Paul K. Faehrmann, Sumeet Khatri, and Jens Eisert. A perturbative gadget for delaying the onset of barren plateaus in variational quantum algorithms. *arXiv preprint arXiv:2210.03099*, 2022.
- [CMP18] Toby S. Cubitt, Ashley Montanaro, and Stephen Piddock. Universal quantum Hamiltonians. *Proceedings of the National Academy of Sciences*, 115(38):9497–9502, 2018.
- [LMF⁺23] Philippe Lewalle, Leigh S. Martin, Emmanuel Flurin, Song Zhang, Eliya Blumenthal, Shay Hacohe-Gourgy, Daniel Burgarth, and K. Birgitta Whaley. A multi-qubit quantum gate using the Zeno effect. *Quantum*, 7:1100, 2023.
- [TRC22] Rahul Trivedi, Adrian Franco Rubio, and J. Ignacio Cirac. Quantum advantage and stability to errors in analogue quantum simulators. *arXiv preprint arXiv:2212.04924*, 2022.

REVIEWER COMMENTS

Reviewer #1 (Remarks to the Author):

My comments from the previous version of this manuscript have been addressed - I would recommend it for acceptance now.

Reviewer #2 (Remarks to the Author):

I thank the authors for the extremely clean and easy to follow presentation of the changes they made to the paper. I believe most of my comments were successfully addressed and would recommend publication of the revised version. Nevertheless, since with the new clarifications I was able to make it further into the proofs, I have a few additional miscellaneous comments I'd be happy if the authors could look at. All equation and theorem numbers below refer to the "Marked up manuscript", where the changes are tracked.

1. Right before (121), it is implied that P' only acts on $O(1)$ systems, but such an assumption is not made in Definition 13.
2. A "tensor identity" is missing after H_{else} in (128).
3. A LaTeX command is visible in (133).
4. Right after (147), I understand H is meant to have no zero eigenvalues instead of H' .
5. Before (150), the additional $O(\eta)$ shift should be $O(\zeta)$, I believe.
6. In (167), the term proportional to ϵ is dropped, but there is no mention of it as there is after (169). I think it would be helpful, together with a small justification.
7. In (168) and (171), should there not be factors of n (resp. n^2 , n^3) arising from the sums over all the S_i, S_j, S_k ?

8. I am still confused about the block diagonality statement right before (178). It seems to imply that H'_i is block diagonal in the $I \otimes P_i$ basis, and that $\exp(-S_i)$ is block off-diagonal in said basis, but I do not see how those follow from S_i being block off-diagonal.

9. Technically there seems to be a factor of 2 missing in the first inequality of (183), due to the commutator.

10. After (187) Δ is said to be $\Theta(nJ)$, but in the statement of the theorem we only assumed the lower bound $\Delta = \Omega(nJ)$

11. In (194), I was expecting the upper bound of the first interval to be $\Delta/2$, instead of $|H| + \omega$.

12. In (201), technically the second term of the RHS should be a remainder, instead of $[X, [X, H']]$ which is just the next order term in the expansion. Also, the final bound is given just as $O(n^3 J' \eta^4)$, while, if I am not mistaken, there is an extra term of order $O(n^2 \eta^3 J)$ which I believe can be dominant if η vanishes very fast.

Reviewer #3 (Remarks to the Author):

I thank the author for their efforts to address my concerns. Nevertheless, I feel still somewhat unsatisfied for a few reasons:

(1) In my original review, I was concerned that in the proposed new definition of "size-independent simulation", there is a requirement that the number of qubits in the simulator must scale only linearly in the target Hamiltonian, which I believe to be unnecessarily strong. (I sympathize with the other requirement that the simulator should have bounded-magnitude interactions.) The authors responded that "If each qubit is encoded into more than a constant number, then it will be impossible to fit them into a space of the same dimension without sacrificing geometric locality." I disagree with this statement, and I point to various results showing that classical problems with nonlocal connectivity can be simulated on a geometrically local system in 2D with a quadratic overhead in qubit number

[see e.g., Lechner et al., Sci. Adv. (2015); Nguyen et al., PRX (2023)].

Indeed, the key no-go result by the author (Prop. 10) motivating their new definition fails when one allows for a quadratic overhead in qubits. It is not clear to me that allowing quadratic overhead would sacrifice geometric locality in the more general setting. Although I agree it's a bit strange that quadratic overhead is necessary for noninteracting system under the definition of simulation by Cubitt et al. [11], an easy alternative solution to achieve linear scaling is to only require simulation up to a certain energy cutoff (e.g. as in Aharonov et al. [18]). I wish the author would be more forthcoming in the main text about the caveats of their result in the main text, which so far has not even mentioned their requirement of linear overhead.

I think a more clear-cut motivation for the authors' new definition is that the previously definition provably does not allow simulation of a general quantum system with a finite-dimensional system with bounded-magnitude interaction, as shown in Aharonov et al. [18], even allowing arbitrary polynomial overhead in the qubits. See also related work by Zhou & Aharonov (2021) which may warrant inclusion with the cited universality results. It would be interesting to explore if the proposed dissipative gadget can circumvent that no-go result.

(2) I was also concerned that the proposed new definition might be too broad to be useful, which can also include digital quantum simulators and Hamiltonian simulators of quantum circuits (e.g., quantum walk and HQCA). I agree with the author's response that requiring that simulating channel T_t to not have feed-forward measurements and to only take $O(t)$ time-complexity to implement, would be sufficient to differentiate it from the aforementioned quantum simulation methods. (I think the paper could also cite the HQCA-based simulator by Bohdanowicz & Brandao (2017).) However, I am not sure if I agree with the requirement of "a static observable encoding \mathcal{E}_{obs} ", since what is considered "static" changes depending on one's frame of reference; it's also not difficult to adjust your measurement device depending on the timestamp of the observable of interest. Perhaps these should be included more formally in the definition to make it more useful for follow-

up works.

(3) As Reviewer 1 and I pointed out, one concern about the paper's new proposed definition and dissipative gadgets is that they might lack a key benefit of the previous definition of analog Hamiltonian simulation -- its robustness against noise without error-correction. I appreciate the authors' effort to include discussion about the mapping of local noise channels, and sympathize with the difficulty to develop more quantitative arguments about the error propagation and robustness. If there is indeed a barrier for dissipative gadgets to achieve scalable and error-robust implementation, I think this deserves a couple sentences in the main text so as to be fully transparent about the relative merits and caveats compared to previous definitions of analog quantum simulations.

(4) I still don't understand the "civil engineering analogy," and I find the added phrase makes it more convoluted. My intuition is that one could fix a certain precision of "the building blocks" until the models one wants to simulate become larger and more complex, at which point one would upgrade the precision. Also, this is apparently only used to motivate the requirement of bounded-magnitude interaction, not the requirement of linear overhead in qubits that I have been concerned about.

In conclusion, while I like the paper and would support its eventual publication in some form, I wish the authors could address the above concerns better to clarify the important caveats of their results before publication.

Dear Referees,

Thank you for the additional comments on the revised manuscript. We have made further edits which we hope address these concerns. The third version of the manuscript is attached, as well as a version with edits highlighted, and a list of changes as before. Below are the reviewer comments, with our responses in blue.

Reviewer 1

My comments from the previous version of this manuscript have been addressed — I would recommend it for acceptance now.

Reviewer 2

I thank the authors for the extremely clean and easy to follow presentation of the changes they made to the paper. I believe most of my comments were successfully addressed and would recommend publication of the revised version. Nevertheless, since with the new clarifications I was able to make it further into the proofs, I have a few additional miscellaneous comments I'd be happy if the authors could look at. All equation and theorem numbers below refer to the "Marked up manuscript", where the changes are tracked.

We are very grateful to the referee for their thorough and careful reading of the proofs.

1. Right before (121), it is implied that P' only acts on $O(1)$ systems, but such an assumption is not made in Definition 13.

The fact that P' acts on $O(1)$ systems comes from the assumptions of Theorem 16. We have now added some text to clarify this.

2. A "tensor identity" is missing after H_{else} in (128).

This has now been corrected.

3. A LaTeX command is visible in (133).

We have now fixed this too.

4. Right after (147), I understand H is meant to have no zero eigenvalues instead of H' .

Indeed, this was a typo; it should have been H . This has now been fixed.

5. Before (150), the additional $O(\eta)$ shift should be $O(\zeta)$, I believe.

This was also a typo, and has now been fixed.

6. In (167), the term proportional to ϵ is dropped, but there is no mention of it as there is after (169). I think it would be helpful, together with a small justification.

A few words have now been added to justify this step. Specifically, since Proposition 22 is a statement about the limit of small ϵ and η , we can neglect the terms which will be dominated in this limit.

7. In (168) and (171), should there not be factors of n (resp. n^2, n^3) arising from the sums over all the S_i, S_j, S_k ?

The absence of these factors is a consequence of the assumptions on locality. Since H_i is local, only $O(1)$ of the terms in $S = \sum_j S_j$ will not commute with H_i . Hence $[S, H_i]$ consists of only $O(1)$ terms, and they are local. The same argument again shows that $[S, [S, H_i]]$ avoids the scalings with n too.

As you point out, this was not explained in the manuscript — a couple of sentences have now been added directly below both equations to clarify this.

8. I am still confused about the block diagonality statement right before (178). It seems to imply that H'_i is block diagonal in the $I \otimes P_i$ basis, and that e^{-S_i} is block off-diagonal in said basis, but I do not see how those follow from S_i being block off-diagonal.

Apologies for this, the justification given for the statement was completely incorrect. Here is the correct explanation:

By construction, $P'_i = e^{S_i}(\mathbb{I} \otimes P_i)e^{-S_i}$ is the low-energy projector for H'_i , so H'_i is block diagonal with respect to P'_i .

Equivalently, $\tilde{H}_i = e^{-S_i} H'_i e^{S_i}$ is block-diagonal with respect to $(\mathbb{I} \otimes P_i)$, just by rotating everything by e^{S_i} . Since S_i is the generator of the Schrieffer-Wolff transformation, it is off-diagonal with respect to both bases — but e^{-S_i} is not necessarily off-diagonal.

The explanation in the text has been edited accordingly.

9. Technically there seems to be a factor of 2 missing in the first inequality of (183), due to the commutator.

The factor of 2 has now been added.

10. After (187) Δ is said to be $\Theta(nJ)$, but in the statement of the theorem we only assumed the lower bound $\Delta = \Omega(nJ)$

This has now been corrected — the lower bound is sufficient for the proof.

11. In (194), I was expecting the upper bound of the first interval to be $\Delta/2$, instead of $\|H\| + \omega$.

This was indeed a typo, it has now been fixed.

12. In (201), technically the second term of the RHS should be a remainder, instead of $[X, [X, H']]$ which is just the next order term in the expansion. Also, the final bound is given just as $O(n^3 J' \eta^4)$, while, if I am not mistaken, there is an extra term of order $O(n^2 \eta^3 J)$ which I believe can be dominant if η vanishes very fast.

The remainder is bounded using (116), so that the next term of the expansion can be used — text has been added just above the equation to clarify this.

The $O(n^2 \eta^3 J)$ term has been added to the equation (luckily this doesn't affect the final result of the proposition since the condition (46) ensures it will be dominated by the $O(n\eta J)$ term in ϵ').

Reviewer 3

I thank the author for their efforts to address my concerns. Nevertheless, I feel still somewhat unsatisfied for a few reasons:

- (1) In my original review, I was concerned that in the proposed new definition of “size-independent simulation”, there is a requirement that the number of qubits in the simulator must scale only linearly in the target Hamiltonian, which I believe to be unnecessarily strong. (I sympathize with the other requirement that the simulator should have bounded-magnitude interactions.) The authors responded that “If each qubit is encoded into more than a constant number, then it will be impossible to fit them into a space of the same dimension without sacrificing geometric locality.” I disagree with this statement, and I point to various results showing that classical problems with nonlocal connectivity can be simulated on a geometrically local system in 2D with a quadratic overhead in qubit number [see e.g., Lechner et al., *Sci. Adv.* (2015); Nguyen et al., *PRX* (2023)].

Sorry for the imprecise previous statement; the sentence about fitting into a space of the same dimension has now been amended to include a caveat that interaction strengths should also not grow.

The qualitative idea for this statement is that, if each qudit is encoded into $\Omega(n^\alpha)$ qudits, for $\alpha > 0$, then the larger size of the system will necessarily mean that distances between some encoded qudits scale with n (if the dimension remains constant). As a result, to overcome Lieb-Robinson bounds and allow correlations to spread through the encoded system with the same speed as before, one needs to correspondingly scale the interaction strengths. We have added Lemma 11, at the end of the qudit-to-qubit section, which gives a more concrete description of the distance scalings.

The references given are both constructions designed for the case of adiabatic quantum computation, for which (approximate) ground-state finding is the only goal — in this case, the simulation of dynamics is not studied so the slower spread of correlations in the larger system is not important (though perhaps slows down the cooling process). In any case, a couple of sentences below Lemma 11 clarify this distinction.

Indeed, the key no-go result by the author (Prop. 10) motivating their new definition fails when one allows for a quadratic overhead in qubits. It is not clear to me that allowing quadratic overhead would sacrifice geometric locality in the more general setting. Although I agree it’s a bit strange that quadratic overhead is necessary for noninteracting system under the definition of simulation by Cubitt et al. [11], an easy alternative solution to achieve linear scaling is to only require simulation up to a certain energy cutoff (e.g. as in Aharonov et al. [18]). I wish the author would be more forthcoming in the main text about the caveats of their result in the main text, which so far has not even mentioned their requirement of linear overhead.

We hope that the edits above complement Proposition 10 and help to motivate the weaker definition of simulation. In the main text, a sentence has been added to clarify the restrictions of the result.

I think a more clear-cut motivation for the authors’ new definition is that the previously definition provably does not allow simulation of a general quantum system with a finite-dimensional system with bounded-magnitude interaction, as shown in Aharonov et al. [18], even allowing arbitrary polynomial overhead in the qubits. See also related work by Zhou & Aharonov (2021) which may warrant inclusion with the cited universality results. It would be interesting to explore if the proposed dissipative gadget can circumvent that no-go result.

We find the no-go result of [Aharonov, Zhou (2018)] is slightly difficult to use to justify our definition, since it is not clear that our weaker notion of analogue simulation should be able to

capture the extreme case of efficiently simulating any Hamiltonian with all-to-all connectivity — it seems feasible that a no-go result of this kind could apply to a more general notion of simulator too.

We thank the reviewer for pointing out that we are missing a citation to [Zhou, Aharonov (2021)]. This was an oversight, and citations have now been added in the main text (with the other other referneces after “many so-called universal families”) and the supplementary material (below the definition of analogue quantum simulation).

- (2) I was also concerned that the proposed new definition might be too broad to be useful, which can also include digital quantum simulators and Hamiltonian simulators of quantum circuits (e.g., quantum walk and HQCA). I agree with the author’s response that requiring that simulating channel T_t to not have feed-forward measurements and to only take $O(t)$ time-complexity to implement, would be sufficient to differentiate it from the aforementioned quantum simulation methods. (I think the paper could also cite the HQCA-based simulator by Bohdanowicz & Brandao (2017).) However, I am not sure if I agree with the requirement of “a static observable encoding \mathcal{E}_{obs} ”, since what is considered “static” changes depending on one’s frame of reference; it’s also not difficult to adjust your measurement device depending on the timestamp of the observable of interest. Perhaps these should be included more formally in the definition to make it more useful for follow-up works.

We were slightly wary of including time-dependent \mathcal{E}_{obs} in the definition since it is not easy to rule out the complexity of the simulation task being hidden in this decoding — and the time-dependence might in theory be transferred into the simulation channel T_t . Nevertheless, this is an insightful comment and a remark to this effect has been added to the supplementary information just below the definition of analogue simulation. The statements about quantum walks have also been edited accordingly, and citation has been added to the Bohdanowicz-Brandao paper.

- (3) As Reviewer 1 and I pointed out, one concern about the paper’s new proposed definition and dissipative gadgets is that they might lack a key benefit of the previous definition of analog Hamiltonian simulation – its robustness against noise without error-correction. I appreciate the authors’ effort to include discussion about the mapping of local noise channels, and sympathize with the difficulty to develop more quantitative arguments about the error propagation and robustness. If there is indeed a barrier for dissipative gadgets to achieve scalable and error-robust implementation, I think this deserves a couple sentences in the main text so as to be fully transparent about the relative merits and caveats compared to previous definitions of analog quantum simulations.

A couple of sentences have now been added to the penultimate paragraph of the main text, which we hope clarify these caveats.

- (4) I still don’t understand the “civil engineering analogy,” and I find the added phrase makes it more convoluted. My intuition is that one could fix a certain precision of “the building blocks” until the models one wants to simulate become larger and more complex, at which point one would upgrade the precision. Also, this is apparently only used to motivate the requirement of bounded-magnitude interaction, not the requirement of linear overhead in qubits that I have been concerned about.

The idea was that it was not the precision of building blocks that posed a problem, but their magnitude — but we agree that this analogy was perhaps unnecessarily convoluted, and it has now been removed.

In conclusion, while I like the paper and would support its eventual publication in some form, I wish the authors could address the above concerns better to clarify the important caveats of their results before publication.

REVIEWERS' COMMENTS

Reviewer #2 (Remarks to the Author):

The authors have satisfactorily replied to my comments and thus I would recommend publication. I have three additional minor comments:

1. After the newly introduced Lemma 11, the authors derive a consequence "In particular (...) there exists a pair of sites (...) whose encodings have...". I would say that statement and the one from the lemma are far enough apart that the reasoning feels abrupt and could use a justification. I expected an argument of the form: considering a point x_i , counting all x_j at a distance $n^{(1-\lambda)/D}$, then all X_j at a distance $O(n^{1/D})$ of X_i , then showing that the former is larger than the latter, so for a pair (x_i, x_j) , (X_i, X_j) are further apart than $n^{1/D}$

2. In (182), second line, the quadratic term in S_i seems to not be needed given we have the remainder R_i .

3. In (186), ω is defined with three terms, and this would be a would spot to clarify if one dominates and we are going to drop the others. I interpret, but I do not know if I am right, that the last term is dropped always, as ϵ is $o(1)$ while J is $\Theta(1)$. Later in (197) ω/Δ is taken to be $O(n\eta^2)$, while the term $O(n\eta J/J')$ is dropped and we are not told how these two compare, other than both being $o(1)$.

Reviewer #3 (Remarks to the Author):

The authors have addressed my concerns about the previous version of the manuscript satisfactorily. I would recommend it for publication.

Dear Referees,

We would like to thank all three anonymous reviewers once again for their very helpful and insightful comments on the work. Below are the remaining comments, which we have addressed with our responses in blue.

Reviewer 2

The authors have satisfactorily replied to my comments and thus I would recommend publication. I have three additional minor comments:

1. After the newly introduced Lemma 11, the authors derive a consequence “In particular (...) there exists a pair of sites (...) whose encodings have...”. I would say that statement and the one from the lemma are far enough apart that the reasoning feels abrupt and could use a justification. I expected an argument of the form: considering a point x_i , counting all x_j at a distance $n^{(1-\lambda)/D}$, then all X_j at a distance $O(n^{1/D})$ of X_i , then showing that the former is larger than the latter, so for a pair (x_i, x_j) , (X_i, X_j) are further apart than $n^{1/D}$.

A couple of additional sentences have been added to this paragraph to explain the argument.

2. In (182) (now SI equation 128), second line, the quadratic term in S_i seems to not be needed given we have the remainder R_i .

This has now been corrected.

3. In (186) (now SI equation 132) ω is defined with three terms, and this would be a would spot to clarify if one dominates and we are going to drop the others. I interpret, but I do not know if I am right, that the last term is dropped always, as ϵ is $o(1)$ while J is $\theta(1)$.

A sentence has been added to clarify this.

Later in (197) (now SI equation 143) ω/Δ is taken to be $O(n\eta^2)$, while the term $O(n\eta J/J')$ is dropped and we are not told how these two compare, other than both being $o(1)$.

An extra line of detail has been added to the equation to explain this. The $O(n\eta^2)$ term dominates because

$$\begin{aligned}\frac{\omega}{\Delta} &= O(n\eta^2 + n\eta J/J' + n\eta\epsilon/J') \\ &= O\left(n\eta^2\left(1 + \frac{J}{\eta J'} + \frac{\epsilon}{\eta J'}\right)\right) \\ &= O(n\eta^2(1 + o(1/n) + o(\epsilon/nJ))) \\ &= O(n\eta^2),\end{aligned}$$

where in the third line the assumptions of the theorem are used.

Reviewer 3

The authors have addressed my concerns about the previous version of the manuscript satisfactorily. I would recommend it for publication.